# Optimizing cabin air inlet velocities and personal risk assessment: Introducing the Personal Contamination Ratio (*PCR*) method for enhanced aircraft cabin infection risk evaluation

**Renquan Tu[1], Yidan Shang**📷[2]*****, Xueren Li**📷[3]**, Fajiang He[1]*****, Jiyuan Tu[3]**

**1** College of Air Transportation, Shanghai University of Engineering Science, Shanghai, China, **2** School of Mechanical and Automotive Engineering, Shanghai University of Engineering Science, Shanghai, China, **3** School of Engineering, RMIT University, Bundoora, VIC, Australia

* yidan_shang@163.com (YS); mikehfj@sues.edu.cn (FH)

**Data Availability Statement:** All relevant data are within the paper and its Supporting Information files.

## Abstract

Recurrent epidemics of respiratory infections have drawn attention from the academic community and the general public in recent years. Aircraft plays a pivotal role in facilitating the cross-regional transmission of pathogens. In this study, we initially utilized an Airbus A320 model for computational fluid dynamics (CFD) simulations, subsequently validating the model's efficacy in characterizing cabin airflow patterns through comparison with empirical data. Building upon this validated framework, we investigate the transport dynamics of droplets of varying sizes under three air supply velocities. The Euler-Lagrangian method is employed to meticulously track key parameters associated with droplet transport, enabling a comprehensive analysis of particle behavior within the cabin environment. This study integrates acquired data into a novel PCR (Personal Contamination Rate) equation to assess individual contamination rates. Numerical simulations demonstrate that increasing air supply velocity leads to enhanced stability in the movement of larger particles compared to smaller ones. Results show that the number of potential infections in the cabin decreases by 51.8% at the highest air supply velocity compared to the base air supply velocity, and the total exposure risk rate reduced by 26.4%. Thus, optimizing air supply velocity within a specific range effectively reduces the potential infection area. In contrast to previous research, this study provides a more comprehensive analysis of droplet movement dynamics across various particle sizes. We introduce an improved method for calculating the breathing zone, thereby enhancing droplet counting accuracy. These findings have significant implications for improving non-pharmacological public health interventions and optimizing cabin ventilation system design.

**Funding:** This research was funded by the National Natural Science Foundation of China (Grant No. 82370101) and the Program for Professor of Special Appointment (Eastern Scholar) at Shanghai Institutions of Higher Learning (Project ID: 0920000016). The funders had no role in study design, data collection and analysis, the decision to publish, or the preparation of the manuscript.

**Competing interests:** NO authors have competing interests.

# 1. Introduction

Airborne infectious diseases pose a significant threat to global public health, with the potential to spread rapidly and overwhelm healthcare systems during outbreaks. These respiratory-borne illnesses can lead to widespread infections, affecting millions of people worldwide each year. The transmission of such diseases has become a focal point of research, particularly in light of recent global health crises [1–3]. Among the various airborne infectious diseases, some stand out due to their impact and prevalence. For instance, Respiratory Syncytial Virus (RSV) spreads extensively during certain periods each year, with a median duration of 4.6 months, affecting millions globally [4]. More recently, the COVID-19 pandemic has had an unprecedented global impact, resulting in 775.69 million confirmed cases and 6.95 million deaths [5]. This pandemic has significantly raised awareness about the transmission of respiratory diseases and led to extensive research and vaccination campaigns [6–8]. The emergence of influenza during the later stages of the COVID-19 pandemic further highlights the threat posed by respiratory diseases. These outbreaks have intensified the focus on transmission studies, revealing the potential for rapid spread and severe strain on healthcare resources. In the current globalization period, air travel in particular has become an essential way of linking people worldwide. In this mode of transportation, air cabin plays an important role in spreading infectious diseases [9–12]. Hence, it is vital to create ways to understand how infectious diseases spread within aircraft cabins under different circumstances. Prioritizing research in this area is essential to protecting public health and preventing future outbreaks.

In previous indoor air quality studies, researchers have mostly focused on simulating how droplets spread in closed spaces with either natural airflow or mechanical ventilation systems [13]. These spaces include, but are not limited to, environments such as aircraft cabins, subway, and supermarkets [14–16]. Among these, the study of cabin ventilation systems has attracted considerable attention from scholars due to their unique characteristics and widespread application. This research focus has a notable example in Li et al. [17], who constructed an actual cabin environment to quantify airflow in three distinct directions. Employing large-scale 2D particle image velocimetry (PIV), they compared their experimental findings with computational fluid dynamics (CFD) numerical simulations. This endeavor offered insights into the airflow dynamics within cabin models. Building on this foundation, Cao et al. [18] investigated further into the domain of droplet transmission within cabin environments, considering droplets of varying diameters under different ventilation systems. This study reviewed existing literature data, highlighting the advantages and disadvantages of various parameters. Taking a substantial step forward, Yan et al. [19] built upon the experimental data from Li et al. [17] to validate CFD-simulated airflow patterns within cabin spaces. Notably, this pioneering study introduced the concept of a spherical breathing zone aimed to statistically analyze pollutant concentrations in a defined area, including critical infection probability data. This specific diameter choice aligns with guidelines from the World Health Organization and Safe Work Australia [20–22], which recommend a 30cm range for assessing the breathing zone.

However, Yan et al. [19]'s investigations predominantly focused on droplets of a representative size, which may not be widely applicable to all scenarios. Such an approach falls short in providing comprehensive insights into the diverse spread trajectories and characteristics of droplets of varying sizes. According to the existing measurement and analysis results [23–26], the droplets discharged by the human respiratory system are composed of droplets of different sizes, which are usually categorized into two scenarios. One is that the larger droplets quickly deposit on the ground under the effect of gravity, and another is that the smaller droplets spread with the airflow and gradually deposit at a distance from the discharge outlet. Thus, it is important to consider the influence of the different diameters on the spread trajectories.

The advent of 2020 ushered in a global upheaval with the onset of the COVID-19 pandemic, prompting a surge in research pertaining to the transmission of respiratory droplets within enclosed spaces, particularly aircraft cabins. Kong et al. [27] undertook a significant initiative by simulating the cabin of a commercial wide-body airliner. His groundbreaking work involved an enhancement of the Wells-Riley equation to discern the intricacies of droplet diffusion, thus establishing a direct correlation between droplet dispersion and the ventilation system. This study introduced innovative ventilation systems capable of mitigating infection risks, subsequently offering recommendations for cabin designers. Further, Zee et al. [28] engineered a simulated control group within cabin scenarios. By manipulating airflow rates and initial conditions and observing airflow patterns, droplet diffusion extents, and lifespans within the cabin environment, this method enabled the practical calculation of infection probabilities for passengers. It can be seen from the above that in the research on the spread of droplets in the cabin, the research focus is mainly on how droplets spread with the airflow in the cabin, while there is less research on the spread of droplets near the breathing area of the head of passengers in the cabin. Recognizing the need to address the gaps in research concerning droplet dispersion near passengers' respiratory zones, Kuga et al. [29] proposed an innovative method to identify stable breathing zones in transient conditions. Their approach accounted for various factors, including passenger postures and air re-inhalation. Furthermore, Abouelhamd et al. [30] extended this method's application to semi-outdoor settings, examining both steady and transient states. Their research involved simulations across eight wind directions and four speeds, culminating in a revised breathing zone range under different Scale for Ventilation Efficiency 5 (SVE5) values. Notably, this refined methodology offered a more accurate assessment of infection risk by narrowing the scope of the respiratory zone and enhancing breathing zone calculations for broader research purposes. While elucidating breathing zone dynamics is essential, quantifying infection risk remains equally pivotal in indoor pollutant analysis. In this context, Zhang et al. [31] presented a susceptible exposure index that quantified passenger infection risk at various locations but overlooked the critical influence of air supply. Cough droplets, acting as vectors for viral transmission, can be transported through airflow and inhalation. The omission of air supply in such analyses can lead to substantial inaccuracies. To address this, the Wells-Riley model was introduced as a simplified yet influential framework that predicted infection probabilities based on room ventilation and respiratory airflow. Sun et al. [32] not only considered factors that influence the result of the simulation but also proposed a perfect-mixing-based Wells-Riley model [33] was modified by introducing the social distancing index and ventilation effectiveness. They display infection probabilities in various scenarios but neglect the cabin environment's infection risk. Then Shang et al. [34] took water vapor evaporation into consideration in the simulation, which improved the realism and accuracy of the simulation. Sun et al. [32] made further improvements to the Wells-Riley model. A model was constructed to predict infection risk in offices and suggest social distancing.

In these existing studies, most studies focus on exploring the spread of droplet in the indoor environment, without specific analysis of droplet dispersion in the breathing zone. However, the study on the droplet distribution in the breathing zone only compares a more accurate range of the breathing zone, without combining the outside of the breathing zone, Therefore, it is necessary to combine breathing zone with practical problems. In this study, we've developed an advanced computational fluid dynamics (CFD) model to predict how viral droplets spread in aircraft cabins. Our model includes a detailed 3D representation of the cabin, considering crucial factors such as air supply, exhaust locations, simplified manikins, and seating arrangements. We've used real measurements from aviation flight simulator data to account for various variables, including gravity, temperature dynamics, air supply patterns, and

passenger breathing rates. Our simulations analyze cabin airflow, the path of cough-generated droplets, and how expelled droplets move in three different air supply scenarios. Improved accuracy by calibrating our model using a modified Wells-Riley model, providing more predictable Pathogen Index (*PI*) predictions in various situations. In addition, we innovatively proposed *PCR*(*Personal comtamination ratio*) to predict personal contamination ratios to more intuitively display infection risk and exposure risk distribution. We've also adopted a novel breathing zone definition proposed by Abouelhamd et al. [30]. Additionally, we've introduced a crucial metric: the ratio of droplets within the breathing zone to total expelled droplets, which indicates passenger infection risk in various cabin areas. Our study compares infection probabilities for passengers in different air supply scenarios and examines the impact of seat placement on droplet spread. Our primary aim is to prevent future disease outbreaks, minimize disease transmission risk, and offer valuable guidance to aircraft cabin designers and disease control agencies. Additionally, we aim to assist aircraft manufacturers in enhancing ventilation systems for a cleaner, more comfortable passenger environment.

## 2. Methods

### 2.1. Geometry, mesh and computational setups

This study draws on existing literature and field measurements to investigate droplet transmission within the confines of an Airbus A320 aircraft cabin model. Building upon previous literature [19,27,28,35,36], to optimize computational resources, we have chosen to focus our analysis on a representative subset of the cabin section with three rows. This approach, coupled with periodic boundary conditions, enables us to conduct CFD simulations efficiently. This approach, coupled with periodic boundary conditions, enables us to conduct CFD simulations efficiently. Fig 1 illustrates the computational model of this cabin section with passenger. Ventilation inlets are located at the upper portions of the cabin walls and ceilings, while outlets are placed along the lower sections of the cabin walls.

When considering computational simulated person (CSP) models, we have created a CSP model with the similar dimensions as the model in Yan et al. [19] (as depicted in Fig 1). Following the recommendations of Kong et al. [27], we have represented the CSP model's particle injection, located on the face, as a simplified circular inlet with a diameter of 1.24 cm. In accordance with findings from Haselton et al. [37] and Kuga et al. [29], the exhalation zone is identified to span from 27˚ and 33˚. This study set a central value of 30˚ as the particle injection angle. To optimize the release of droplets, we have positioned the injector at row 2, seat C, as this location aligns with existing literature suggesting it as an ideal point for droplet release, significantly enhancing the transmission dynamics of droplets [38].

As shown in Fig 1, passengers and seats are arranged on two sides of the cabin. There are three rows, each accommodating six persons, making a total number of 18 passengers. To achieve mesh independence, this study experimented with five sets of mesh configurations, utilizing total mesh elements of 2.60 million, 3.92 million, 5.20 million, 6.51 million, 7.84 million, respectively. Significant discrepancies in velocity were observed among the groups with grid numbers of 2.60, 3.92, and 5.20 million, while an increase in mesh elements from 6.51 million to 7.84 million revealed negligible deviation in the velocity field, as illustrated in Fig 2. This indicates that a mesh of 6.51 million elements is suitable for simulating airflow within the computer cabin. Therefore, in our simulations employed a mesh grid totaling 6.51 million polyhedral cells within the cross-section of the three-row cabin, achieving a maximum skewness of 0.81. Fig 3 illustrates the detailed local surface mesh and the periodic surface mesh. Specifically, the grid dimension near the head of the Computational simulated person (CSP) was 5 mm, with a mouth opening mesh size of 1 mm. This configuration is designed to effectively

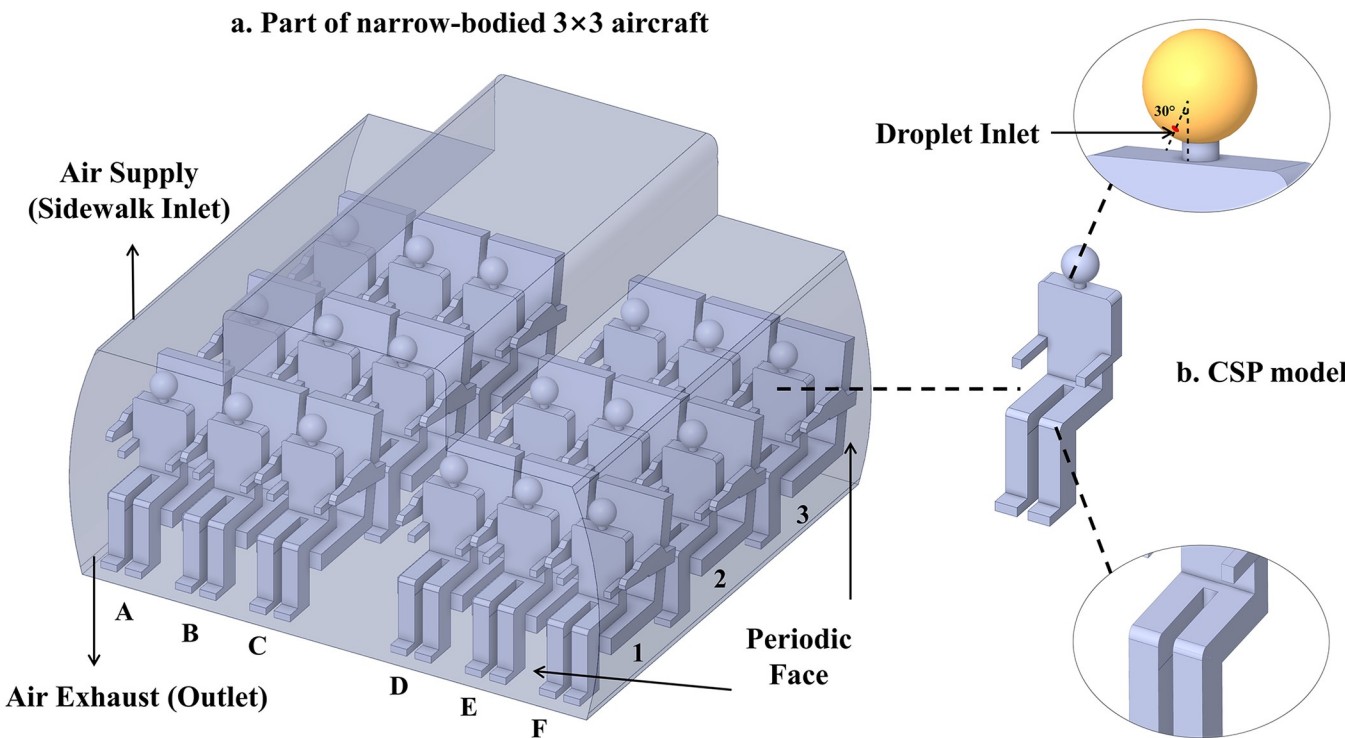

**Fig 1. Computational model of cabin section and passengers.** a: Part of narrow-bodied 3×3 aircraft used to mimic the aircraft cabin environment, restoring the whole cabin through periodic face in post-processing; b: Details of the CSP model. It shows the location of droplet inlet and the geometry details of the model.

capture variations in the airflow patterns and droplet dispersion near the critical breathing zone. The mesh size around the air supply and exhaust areas is maintained at 18 mm to ensure full coverage of these key regions. In order to simulate the airflow within the cabin, this study set a boundary layer mesh with a first height of 0.002m, and a growth rate of 1.2 on the wall. It was ensued that the y+ values were kept below 5.

## 2.2. Boundary conditions setting

The previous literature has highlighted the key role of air flow velocity in the pathogenic dispersion [39]. Previous research has mainly concentrated on pathogenic dispersion in scenario with minimal air supply within cabin environments. It is noteworthy that actually cabin air supply volumes surpass this minimal threshold, as they are established and validated by aircraft designers. Therefore, there is a significant knowledge gap between the diffusion and dispersion of pathogenic under different inlet velocities. To narrow this gap and take account of air supply volume, this study introduced three different velocities to simulate the influence of air supply on droplet trajectories. We have designated the velocity of 0.51 m/s as the baseline inlet velocity at the minimum air supply [40], with 1.03 m/s representing nearly twice the baseline inlet velocity. Initial simulations at both baseline and double baseline velocities showed notable differences in droplet spread. Consequently, an intermediate velocity of 0.77m/s was introduced to address this disparity.

A droplet inlet was set on the head surface of seat C in row 2. This velocity represents the minimum required air supply and the droplet releasing location was selected to simulate the worst-case scenario. Additionally, we have established the inlet and outlet temperatures at 19˚C, in accordance with guidelines in the IATA [41].

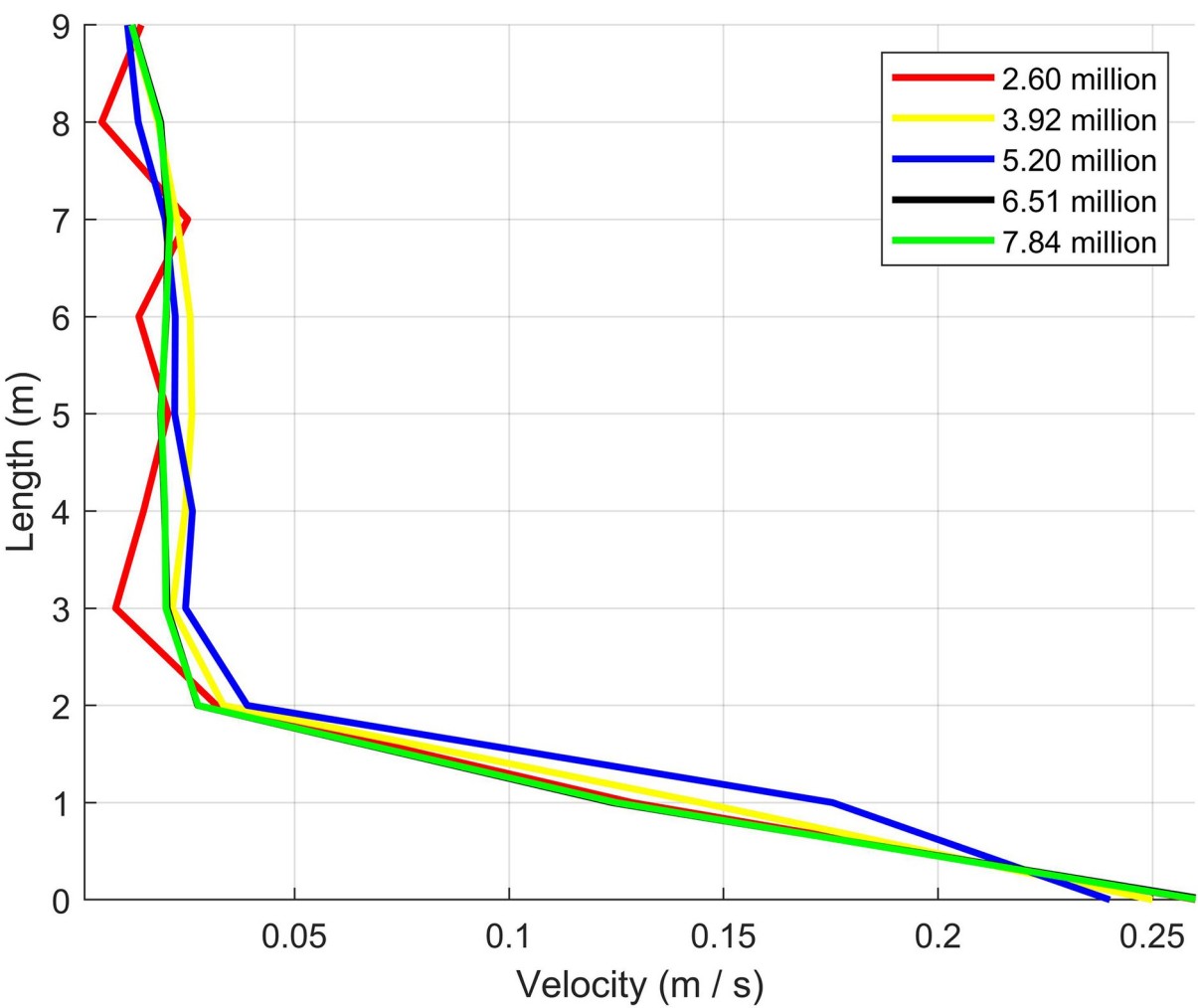

**Fig 2. Mesh independence test based on velocity field of the minimum air supply.**

To simulate the cabin's physical environment, we have set the floor, seats, walls, and ceiling as no-slip surfaces. Additionally, we employed translation as the periodic face type. In terms of the computational simulated person (CSP), we maintained a constant surface temperature of 32˚C [34]. The temperature difference between the CSP and cabin environment induces a natural upward airflow along the CSP's surface due to the buoyancy effect. The flow speed of normal-breath-induced droplets at this inlet is 1 m/s normal to the face, according to the research by Zhang et al. [31]. This study employs the SIMPLE scheme for pressure-velocity coupling and the second-order upwind scheme for momentum space discretization. This combination has been proven particularly suitable for indoor environment simulations, aligning well with the scope of our study [42,43]. Once the flow field stabilizes, we release a total of 100,000 droplets to simulate typical droplet transmission patterns during normal breathing.

### 2.3. Determination of droplets

In this research, we investigate the range of sizes of expelled normal breath droplets, which vary from submicron to several hundred microns in diameter. To simplify our analysis, we have converted the droplet count into a more convenient unit: 'number of droplets per unit

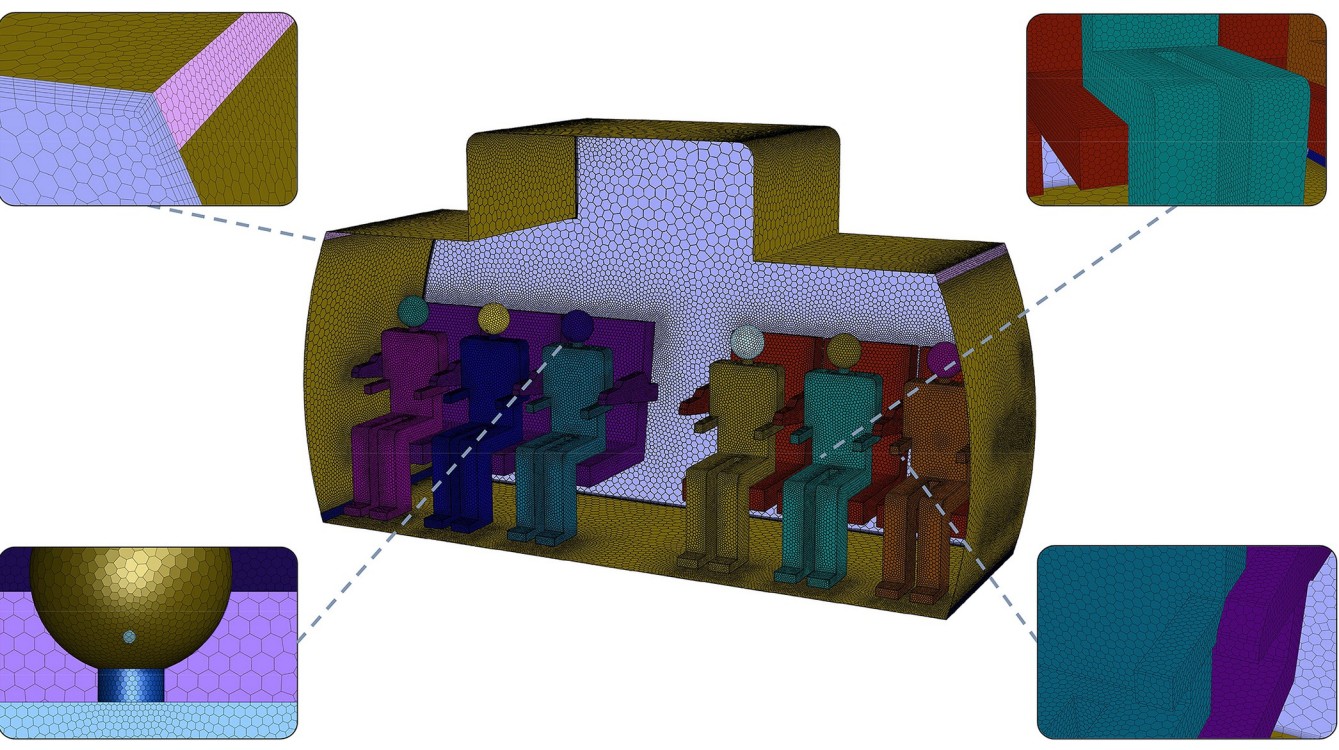

**Fig 3. The surface mesh of the computational model.**

$\mu m$'. According to Atkinson [44], when a man sneeze, talk, or normal breath, they generate various droplets. These droplets can be categorized by size, with those larger than 5 $\mu m$ rapidly falling to the ground due to gravity, on the contrary, the droplets with diameter less than 5 $\mu m$ will float for a while in the air with the airflow. Based on that, 5 $\mu m$ is considered as a cut-off size to distinguish the small and large contaminant. 1 $\mu m$ droplet (small size) and 5 $\mu m$ droplet (large size) was thereupon adopted in this study to investigate the key droplet aerodynamics within the cabin environment. Our primary objective is to investigate the spread characteristics of these two categories of droplets within the cabin environment.

To simulate the droplets based on equivalent aerodynamic principles, we defined them as inert particles, allowing us to focus on their aerodynamic behavior and physical characteristics without involving complex chemical reactions or biodegradation processes. Based on previous research, the droplets are modeled using water-liquid, with a density of 1000 $kg/m^3$. This configuration ensures that we can accurately capture the droplets' behavior in the airflow [45–47].

### 2.4. Governing equations

This study utilized the commercial computational fluid dynamics (CFD) software ANSYS Fluent 2021 R1 for conducting all numerical calculations. The Navier-Stokes(N-S) equations with the Boussinesq approximation were used to simulate airflow field in Eulerian method,

$$\nabla \cdot \overrightarrow{v} = 0 \tag{1}$$

$$(\overrightarrow{v} \cdot \nabla)\overrightarrow{v} = -\frac{\nabla P}{\rho} + \frac{\mu}{\rho}\nabla^2 \overrightarrow{v} + \overrightarrow{g} \tag{2}$$

$$\rho C_p \overrightarrow{u} \cdot \nabla = k\nabla^2 + \rho C_p \overrightarrow{g} \cdot \overrightarrow{u} + Q \tag{3}$$

Where the $\overrightarrow{v}$ is the air velocity vector, $\rho$ is the density of air, $P$ is the static pressure of air, $\overrightarrow{g}$ is the gravity, $\rho$ and $\mu$ are the density and the viscosity of air. $C_p$ is the Specific heat capacity at constant pressure. $T$ is temperature. And $k$ is the thermal conductivity coefficient. $Q$ is the heat source term.

$$\rho \overline{v}_i \frac{\partial \overline{\Phi}}{\partial x_i} - \frac{\partial}{\partial x_i}\left[ \Gamma_{(\varphi, eff)} \frac{\partial \overline{\Phi}}{\partial x_i} \right] = S_\varphi \tag{4}$$

Where $\overline{u}_i$ represents the components of fluid velocity, $\Gamma_{\varphi,eff}$ is the effective diffusion coefficient, $\varphi$ is the flow variables, and $S_\varphi$ represents the source term. Cao et al. [18] has compared that the Realizable k-epsilon, Standard k-epsilon, RNG k-epsilon and SST k-omega. We employed the realizable k-epsilon model for simulating airflow turbulence. From the aspects of velocity, temperature, particle concentration and comprehensive quality, it is found that the Realizable k-epsilon has a better balance between accuracy and computational cost, as highlighted by Li et al. [17]. The particles were tracked by the Lagrangian method. In particles tracking accuracy control, the trapezoidal scheme and analytic discretization scheme were employed to track particle trajectories, with a control tolerance of 1e-05.

$$\frac{d\overrightarrow{u_p}}{dt} = \frac{18\mu}{\rho_p d_p^2}(\overrightarrow{u} - \overrightarrow{u_p}) + \frac{\overrightarrow{g}(\rho_p - \rho)}{\rho_p} \tag{5}$$

Where the $\overrightarrow{u_p}$ is the droplet velocity (m/s), $t$ is the time (s), $\mu$ is the air viscosity (Pa*s), $\rho_p$ is the droplet density (kg/m³), $d_p$ is the droplet diameter (m), $\overrightarrow{u}$ is the air velocity (m/s), $\overrightarrow{g}$ is the acceleration of gravity (m/$s^2$), $\rho$ is the air density (kg/m³),

## 2.5. Modified Wells-Riley model and breathing zone

The Wells-Riley model represents a foundational framework for modeling infectious diseases, initially formulated by Willam F. Wells and Richard L. Riley [33], and widely adopted in the epidemic transmission assessment field [48]. The Probability of Infection (*PI*), defined as the ratio of infected individuals and susceptible individuals within a confined environment, can be approximated by a well-established expression [33],

$$PI = 1 - exp\left( -\frac{Iqpt}{Q} \right) \tag{6}$$

In this equation, $I$ represents the count of infectors, $q$ denotes the quantum generation rate from a single infector, determined through reverse calculations derived from empirical data. $p$ stands for the pulmonary ventilation rate, while $t$ signifies the duration of exposure, and $Q$ represents the room's ventilation rate.

In the classical equation, however, the ventilation modes and social distance scenarios are not taken into account. Sun et al. [32] improved the classical equation through introducing the air distribution effectiveness $E_z$ and social distance index (*SDI*). Among these, the Social Distance Index (*SDI* is also exemplified by $P_d$ in Sun et al. [32]) quantifies the cumulative proportion of droplets expelled through breathing, which ultimately reach the respirable region at a

specified distance $d$, posing the potential risk of inhalation by susceptible individuals.

$$Infection\ Risk = 1 - \exp\left(-SDI\frac{Iqpt}{Q \cdot E_z}\right) \tag{7}$$

In study of Sun et al. [32], the specific breathing zone is not taken into account, which led to the possibility of inaccurate infection risks. Shang et al. [34] modified the *SDI* using a simplistic approach known as the 'Distance-Reaching Method'. This method relied on the assumption that droplets consistently achieved their terminal velocity, balancing drag force, gravity, and buoyancy. Nevertheless, it overlooked droplet height, raising concerns about its accuracy. To address this limitation, Shang et al. [34] demonstrated that only the breathing zone within a radius of less than 30 mm significantly influences droplet inhalation. Thus, the 'Spherical Zone Method' was introduced to calculate the *SDI* more accurately.

$$\begin{aligned} Infection\ Risk\ (With\ Spherical\ Zone\ Method) &= \frac{C}{S} \\ &= 1 - exp\left(-SDI\frac{Iqpt}{QE_Z}\right) \\ &= 1 - exp\left[-SDI\frac{Bqpt}{(Q/N)E_Z}\right] \end{aligned} \tag{8}$$

Afterwards, Kuga et al. [29] introduced a novel concept for the breathing zone, particularly in the context of SVE5. They conducted simulations of various ventilation systems under both steady-state and transient conditions, including displacement ventilation, microgravity ventilation, and mixing ventilation. Their analysis explored the range and characteristics of the breathing zone in diverse scenarios, the proposal of a refined breathing zone concept that offers enhanced accuracy. Compared with the previous sphere breathing zone Shang et al. [34], the later makes the range of breathing zone more improved.

In this study, we built on the previous model and method, adopting the innovative breathing zone concept introduced by Kuga et al. [29] and combining it with the principles of the *PI* to assess infection risk within an aircraft cabin employing a mixing ventilation system. Compared to previous methods used for predicting droplet spread, this study introduces a new exposure risk quantification method which named *PCR* (*Personal Contamination Ratio*). By comparing the *PCR* index within each breathing zone, the concentration of droplets in each breathing zone under macro conditions is displayed straightly, which could clearly contrast the distribution of droplet concentrations. This new method studies the individual exposure risk passengers in different areas rather than simply treating all contacts as infected by default. This is a breakthrough in replacing a global perspective with a local perspective. It is helpful to identify the specific person who is more susceptible to infection among contacts. In addition, this study combines the *PCR* method with Wells-Riley model into a new infection risk assessment model to accurately assess individual relative infection risk.

The defined breathing zone is a rectangular region measuring 4.5 cm in length, 5.4 cm in width, and 21 cm in height, originating from the lower nasal cavity's cross-section, as recommended by Kuga et al. [29]. In view of the simplified human body in the CSP model employed in this study, the starting surface of the breathing zone is defined at the midpoint of the quarter of the semicircular frontal head section, as depicted in Fig 4.

$$Personal\ Contamination\ Ratio = \frac{N_{vs}(V)}{N_{vr}(V)}$$

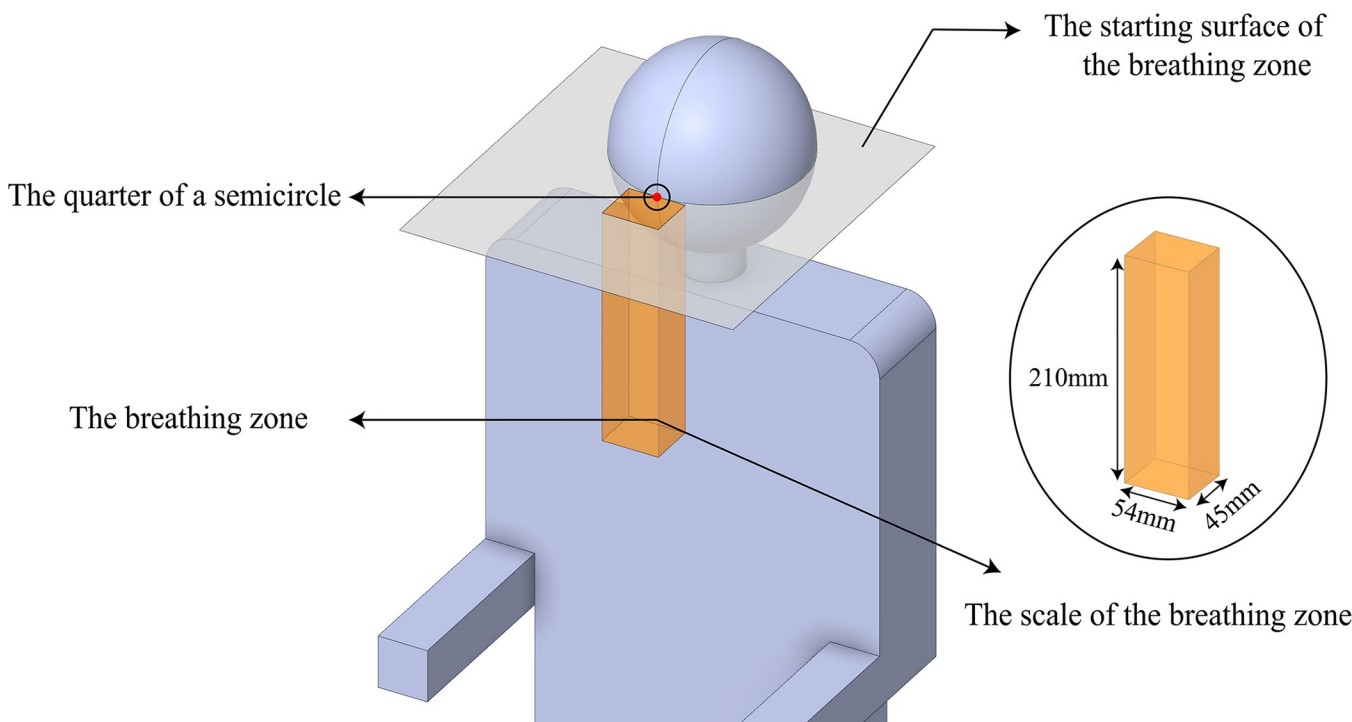

**Fig 4. The scale and location of the breathing zone.**

$$= \frac{\sum_{i(single \; respirable)} V_{cuboid-i} \times C}{V_{cuboid-i} \times C} = \frac{\sum_{i(single \; respirable)} V_{cuboid-i}}{V_{cuboid-i} \times C} \tag{9}$$

Among them, $N_{vs}(V)$ indicates the cumulative number of droplets in a single breathing zone, and $N_{vr}(V)$ represents the cumulative number of droplets particles in all breathing zones. $C$ represents the droplets concentration in the droplets discharged at the moment of discharge, and $d_i$ is the original diameter of the $i$th expelled droplet.

Inserting Eq (1) leads to a refined Wells-Riley model for the regional risk of infection, based on the local particle quantity [49].

$$PI = 1 - \exp\left(-\frac{ln2}{\theta TCID_{50}} \cdot PCR \cdot N_{particles} \cdot p \cdot t\right) \tag{10}$$

Among them, $\theta$ denotes the ratio coefficient of $HID_{50}$ (median infective dose in humans) to $TCID_{50}$, specifically we utilize the influenza data for estimation [49]. $N_{particles}$ represents the number of the particles, $p$ signifies the pulmonary ventilation rate, and $t$ represents the duration of the flight, estimated as 6.5 hours according the data in Sun et al. [32].

## 3. Results and discussion

### 3.1. Model validation

In this study, the experimental measurement conducted by Li et al. [17] is adopted for the purpose of validation. They have significantly contributed by creating a physical air cabin model and employing large-scale 2D Particle Image Velocimetry (PIV) to meticulously measure

airflow patterns across five different sections. The primary measurement areas included beneath the overhead luggage, above the seats, and within the aisle region. These data sets provide velocity information for Rows 3, 4, and 5 in the air cabin, contributing to the validation of the model in advance of simulation. To maintain consistency with experimental conditions, we try to replicate all parameters as closely as possible in order to align with the original experimental setup [17], including air inlet angle and speed, among other variables. We use these velocity measurement data as a benchmark for comparing and validating our simulation results, ensuring the accuracy of our cabin airflow simulations.

In Fig 5, we set six measurement lines in front of the CSP and one in the center of the cabin aisle to capture the velocity data, as indicated in the upper portion of Fig 5. The red line in the lower part of Fig 5 represents the simulation data generated in this study, while the black points represent the experimental data from Li et al. [17]. Lines A, B, E, F, and the Aisle show similar trends when compared to the experimental data. Although lines C and D show deviation within the height range of 1.25 meters to 1.5 meters, while the numerical results demonstrated a satisfactory level of agreement with the corresponding experimental data, especially for heights below 1.25 m, which corresponds to the breathing height for seated occupants. As depicted in Fig 6, in both experimental measurements and numerical simulations, there are similarities in the airflow patterns. After being injected through the air inlet, the airflow converges in the aisle, and experiences intersection and collision, leading to the creation of two symmetrical vortices. The vector raw experimental data is sourced from Li et al. [17].

## 3.2. Airflow field prediction

This study explores the risk of infectious diseases within the aircraft cabin, focusing on the transmission paths of pathogenic. Consequently, this research begins with visualizing cabin airflow characteristics to better understanding of the particle distribution. Representative locations within the cabin would be further identified and sectional analysis points would be established to assess airflow velocity and direction in different cabin sections. This facilitates a thorough investigation into the characteristics of droplet propagation within aircraft cabins, enabling a more comprehensive exploration.

Fig 7 illustrates velocity contour sections in three different directions: cross-section, longitudinal section, and horizontal section. As depicted in the figure, the primary driving force of airflow within the cabin is the air released from the cabin's inlet. The fastest airflow is mainly concentrated along the inner walls of the cabin, while other areas have relatively consistent airflow velocities. Notably, there is the slowest airflow near the aisle. It is evident that the longitudinal and horizontal sections demonstrate comparatively stable airflow velocities.

The difference is the cross sections show significant variations in velocity, indicating a consistent trend across all three sections. The airflow from the inlet descends along the inner walls of the cabin, and upon reaching the floor, some of it rebounds and rises, while the rest escapes through the outlet. In each of the different sections, it is evident that the velocity is higher in the region including the CSP and seats compared to the aisle region. Furthermore, the cabin ceiling region shows relatively stable airflow velocities with minimal fluctuations.

The distribution of particle in the cabin influenced not only by inlet air velocity but also the direction of airflow. As depicted in Fig 8, we present three different perspectives of cabin row: a cross-section, a longitudinal section, and a horizontal section. It is evident that the initial inlet air from the cabin inlet propels forward for a distance before descending along the cabin's sidewalls. Subsequently, the airflow from both the left and right sides meet near the aisle, ascending in opposite directions. Furthermore, the human thermal plume has an influence on the airflow direction as shown in Fig 9.

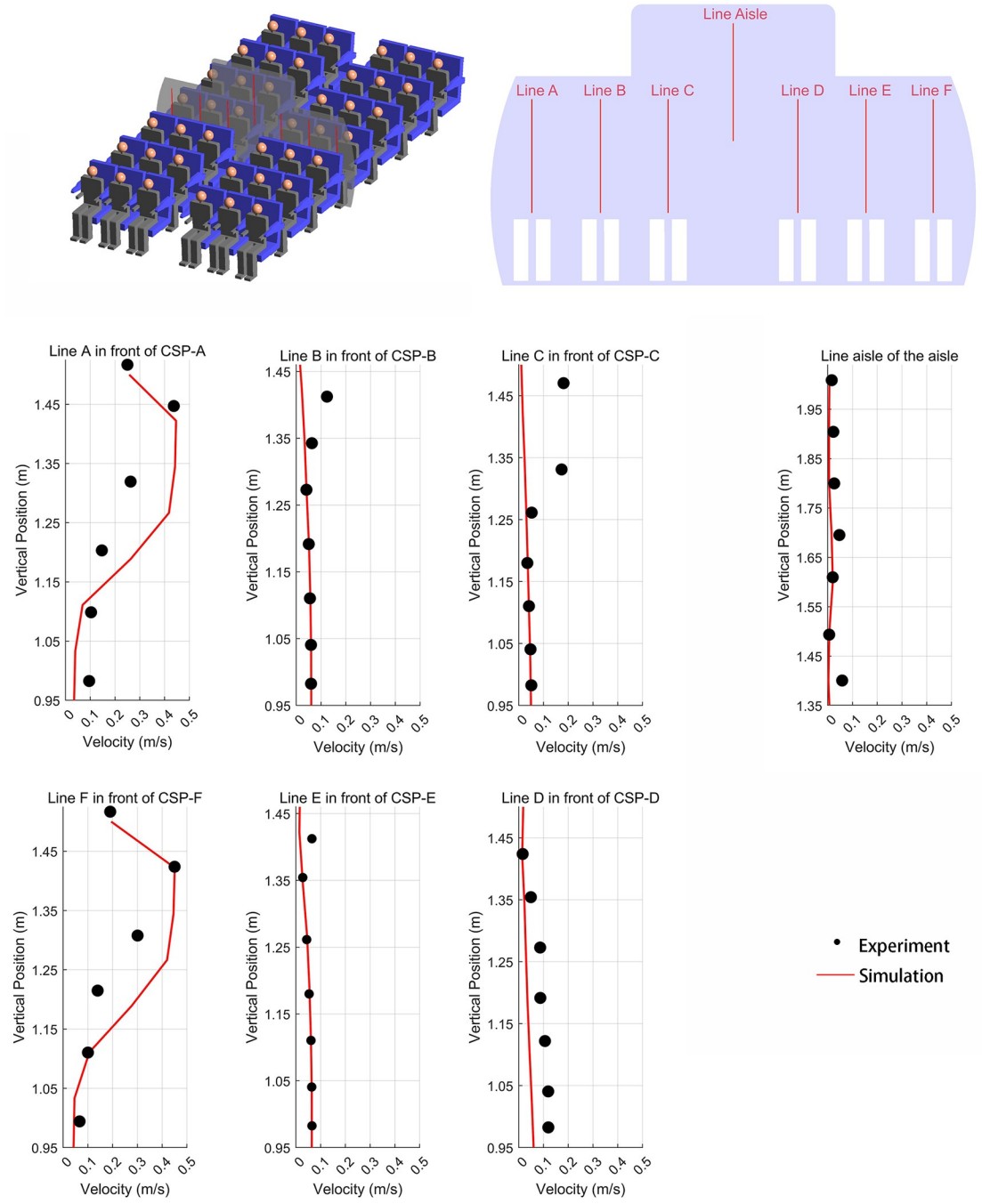

**Fig 5. The comparison of the specific lines in experiment and simulation situation.**

Consequently, as the airflow collides and ascends on either side, it rises to generate two vortices near to the CSP-A and CSP-F regions. Meanwhile, two eddies displayed on the cabin's left and right sides, situated near the ceiling.

Fig 10 shows the velocity counter and vector of the cross section under three inlet speeds. It can be inferred that as the inlet speed increases, the angle between the jet and sidewall gradually increases; In fact, the angle between the sidewall and the ceiling also has a great impact on

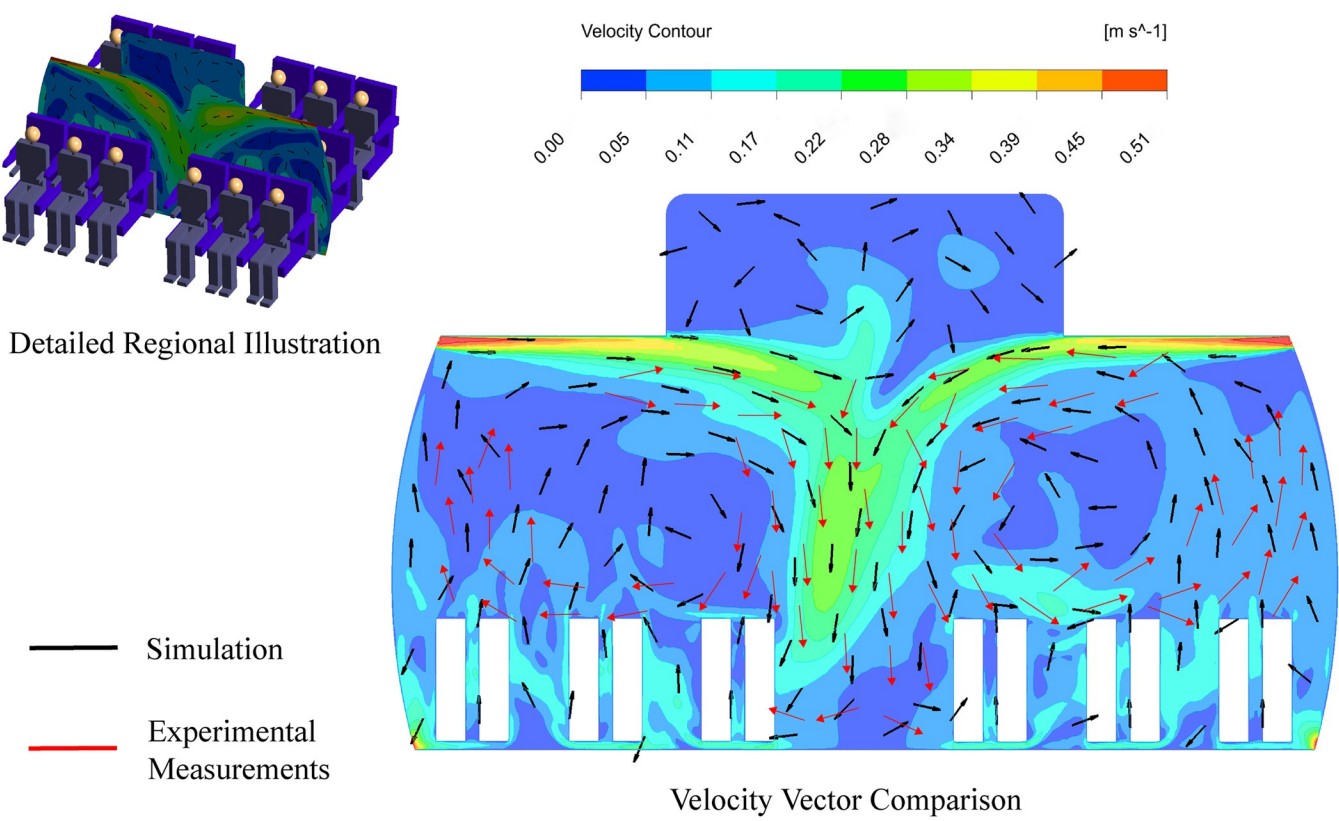

**Fig 6. Comparative analysis of velocity vectors: Simulation and experimental measurements [17] at row 4.**

the path of jet. In short, the overall airflow trend in the cabin seems to have no great influence as the inlet speed increases.

## 3.3. Influence of the ventilation velocity on the transmission path

In this study, there are three cases designed with three different inlet velocity, 0.51 m/s (foundation case), 0.77 m/s, and 1.03 m/s, respectively. These additional cases are based on the foundational air supply case. It's important to note that, due to concerns about comfortable and temperature, we did not simulate scenarios with higher velocities. Nevertheless, the results obtained from the selected velocities provide valuable insights into the relationship between pathogenic and air supply volume.

In the analysis of droplet diffusion paths, we illustrate the trajectories of droplets with sizes of 1 $\mu m$ and 5 $\mu m$ separately in order to demonstrate the impact of different inlet wind velocity on the spread paths. Fig 11 visually shows the trajectories of droplet within the cabin, with trajectory color transitioning as time. Notably, whether considering larger or smaller droplets, their residence times predominantly fall below 350 seconds. Moreover, the direction of their movement opposes the direction of exhalation. The reason of this phenomenon is the meeting of incoming air flows from cabin inlets along the sides, their collision at the aisle's bottom, and subsequent upward movement with the influence of human thermal plumes and respiratory effects. Consequently, the airflow trend propels the droplets towards the rear of the cabin. Therefore, the spread trajectory of droplets is backward, and most of the droplets are distributed behind the injector.

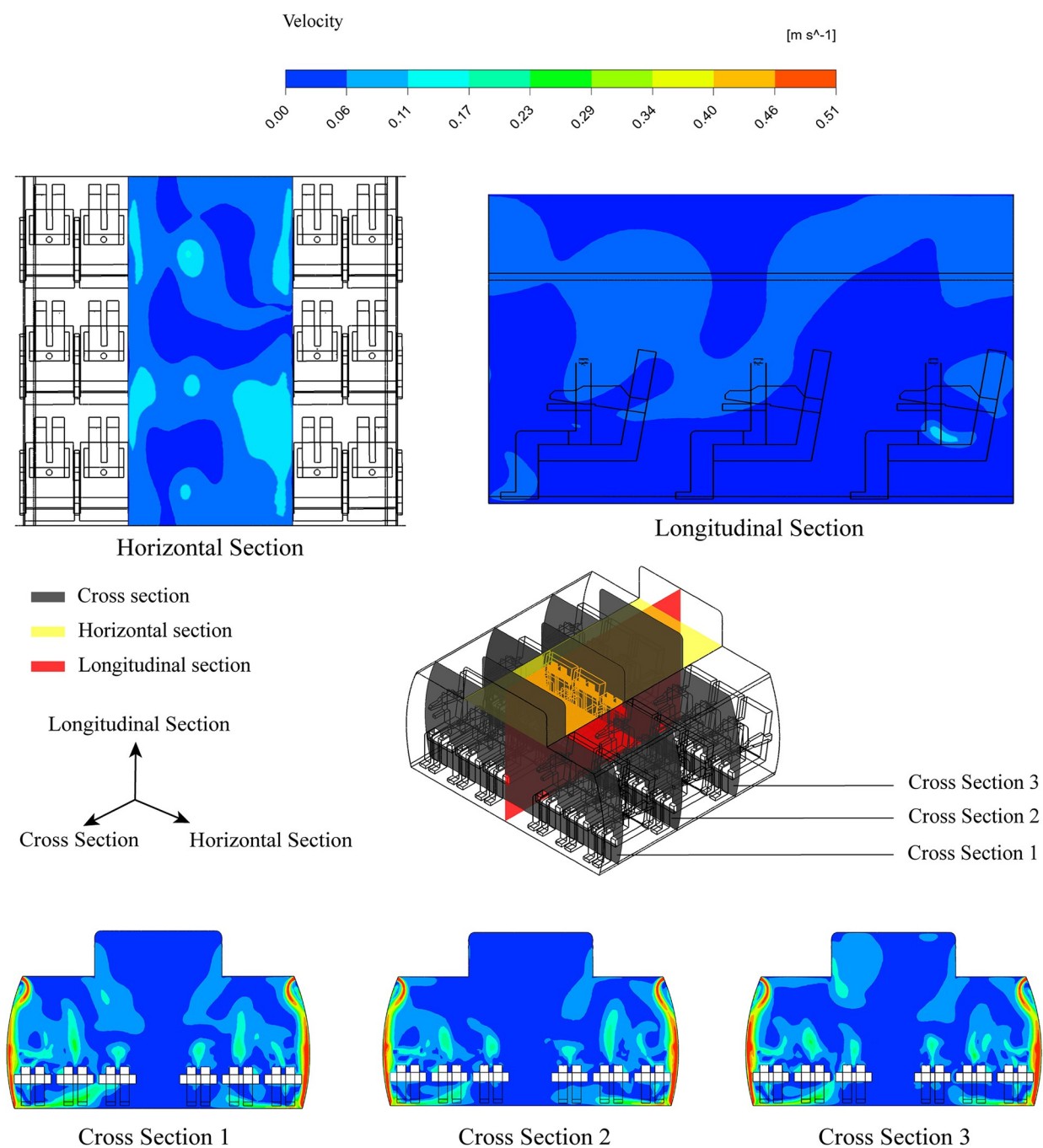

**Fig 7. The velocity contour in all sections of air cabin (0.51 m/s).**

From a detailed perspective, the trajectory of small droplets' spread is directly influenced by the inlet velocity. At the minimum inlet velocity, droplets initially concentrate in two rows behind the injector and gradually disperse to greater distances, including the rear rows of seats in the opposite row. When the inlet velocity is in the mid-speed, droplets diffusion tends to extend to the last few rows of the column, but the spread range is noticeably smaller than at the minimum inlet velocity case. As the velocity is maximum, the droplets spread range shrinks

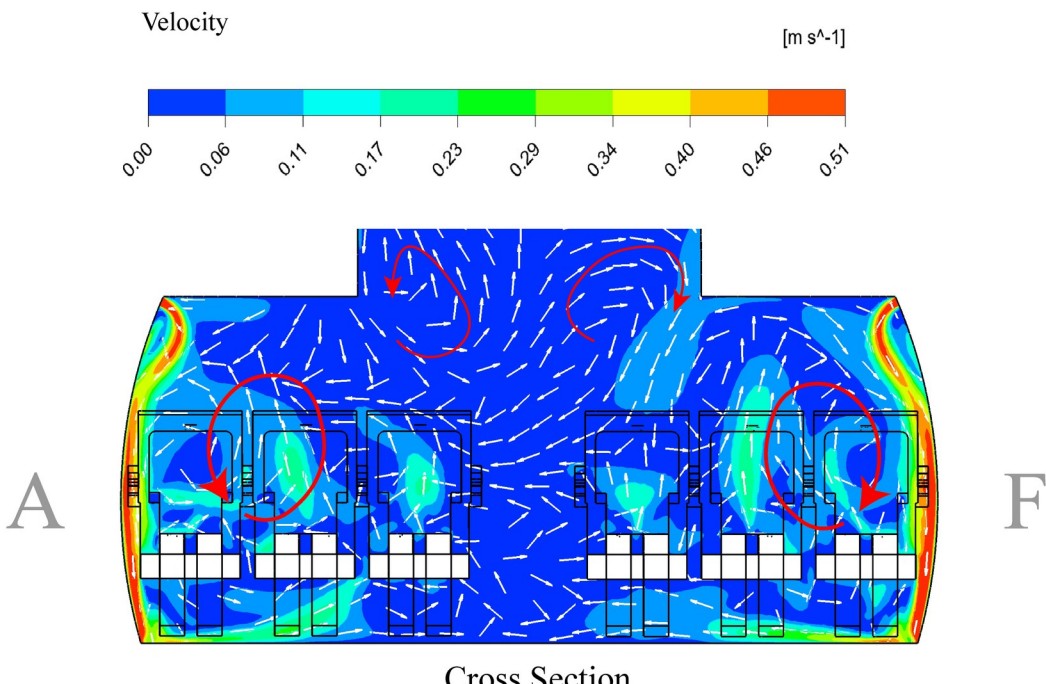

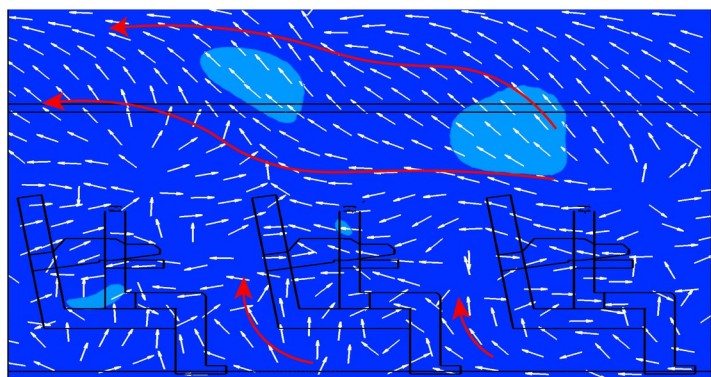

(Row 5: where the infected located)

**Fig 8. The velocity contour and vector in all sections.**

significantly, with large and small droplets following similar trajectories, leading to the rapid escape of many droplets from the cabin.

Conversely, for larger droplets, regardless of whether the inlet air speed is high or low, their motion trajectories are relatively uniform. It's evident that with increasing inlet velocity, the spread distance of larger droplets also increases, resulting in longer residence time compared to other two velocity cases.

In summary, as velocity gradually increases, the diffusion of droplets is somewhat constrained. Smaller, more easily dispersible droplets imply reduced dispersion ranges, while larger droplets do not experience a significant expansion in their dispersion range with increased air flow velocity. However, as the air flow velocity continues to increase, it can

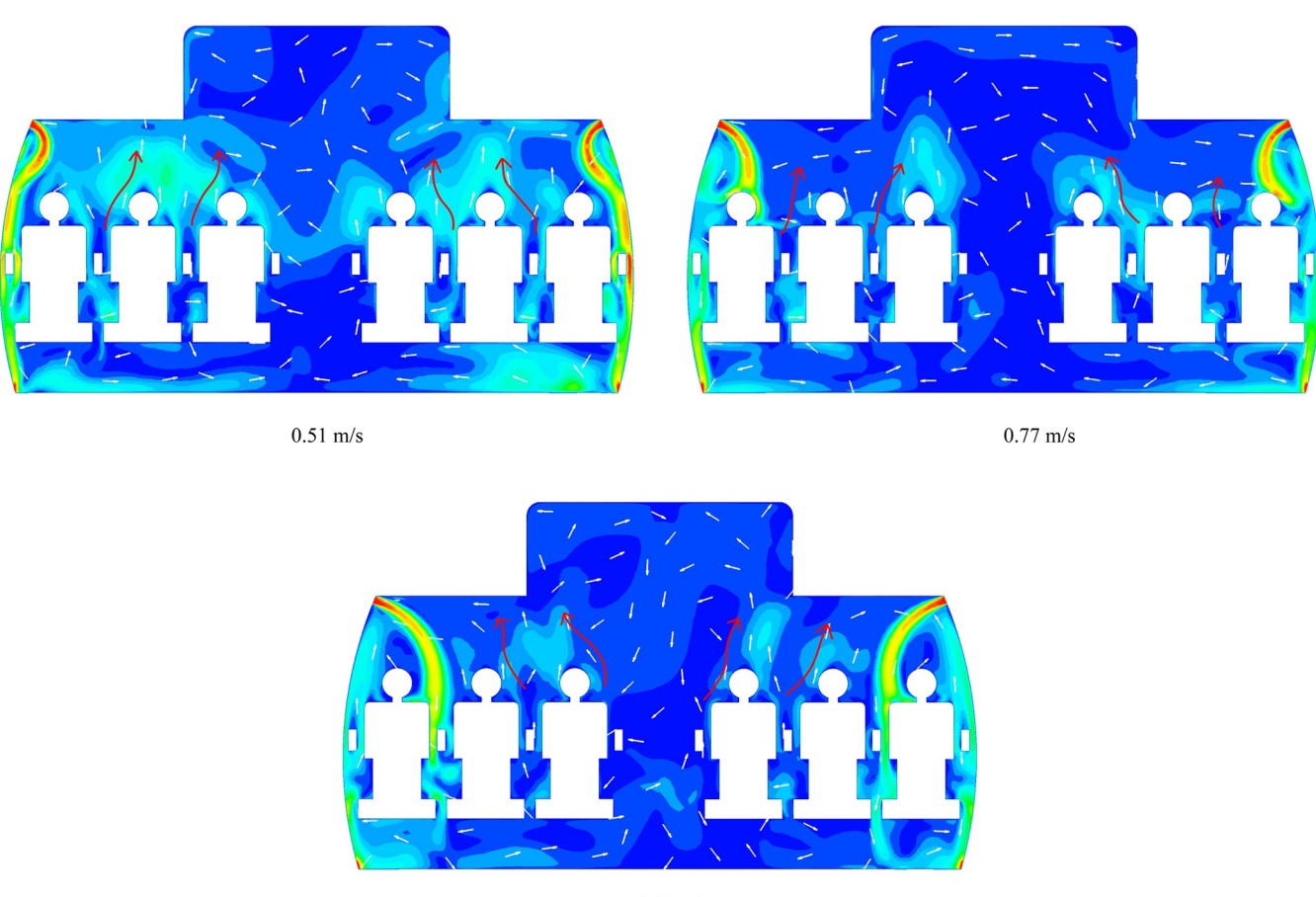

0.51 m/s

0.77 m/s

1.03 m/s

**Fig 9. The velocity contour and vector in all sections.**

induce the movement of droplets of various sizes. While passengers near the injector may initially appear unaffected by droplets, the potential infection area could expand as airflow persists.

### 3.4. Exposure and infection probability in different ventilation velocity

The detailed information of the expelled droplets (e.g., location, residence time, etc.) in each passengers' breathing region was extracted and being further used for exposure and infection quantification using the *PCR* and modified Wells-Riley model To illustrate potentially infected individuals, we create heatmaps and 3D bar chart, and transform the dataset into percentages, enhancing the clarity of infection probabilities. This visual representation shows a direct assessment of variations in infection risk among different passengers, offering insights into the potential extent of infections within the cabin.

Figs 12 and 13 clearly illustrates those individuals at a higher risk of exposure tend to be situated behind the row of injectors. The closer a passenger is to the source of contamination, the more likely they are to be included in the potential infection area. When the inlet velocity changes from its minimum to maximum values, we observe a concentration of high-risk individuals predominantly in rows positioned behind the injector. In an intermediate velocity case, high-risk individuals are more evenly distributed within the cabin. Furthermore, at this intermediate velocity, the potential infected persons appear to be surrounding the Source as a whole.

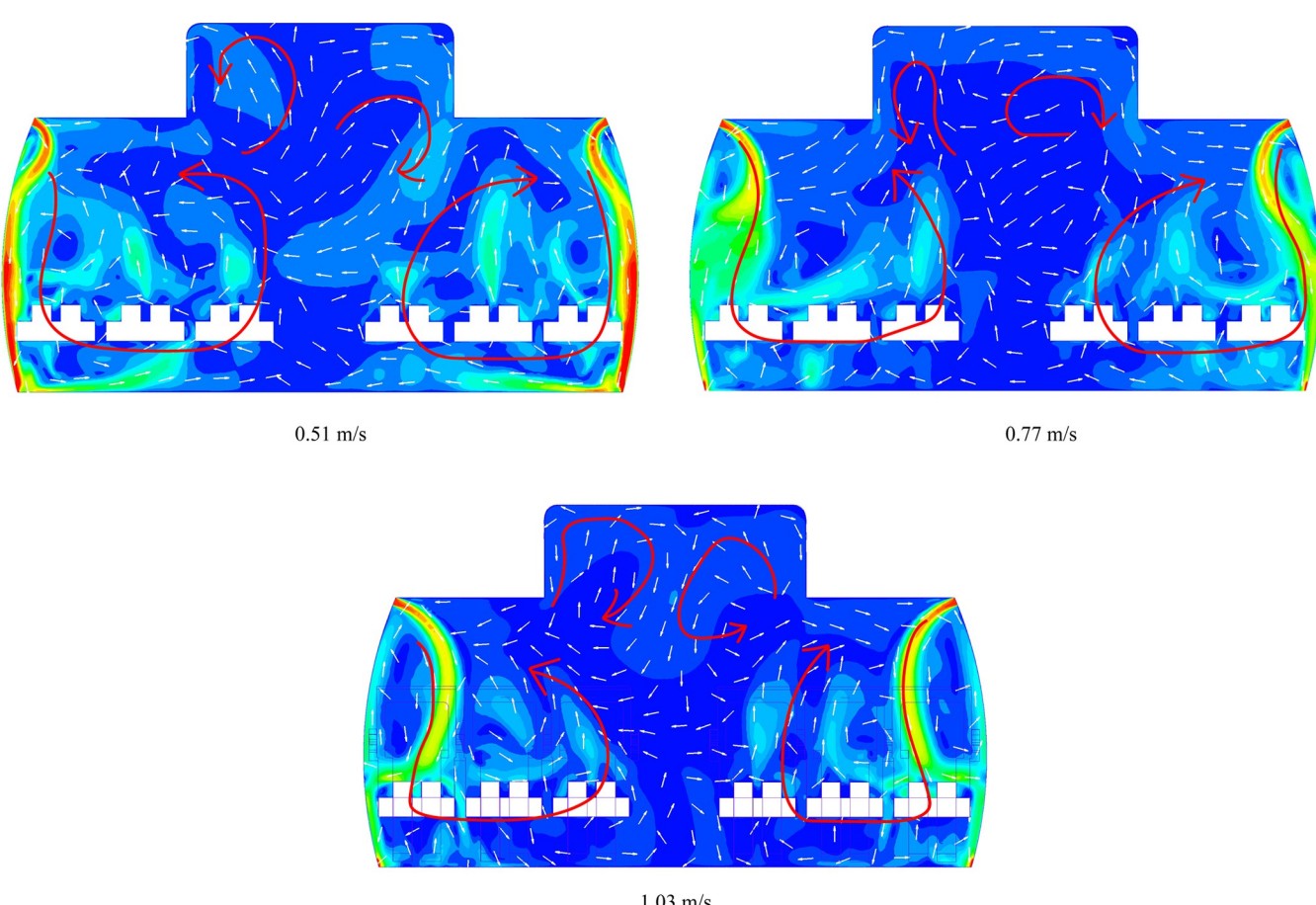

0.51 m/s

0.77 m/s

1.03 m/s

**Fig 10. The velocity contour and vector of three cases with different velocity.**

While a superficial analysis of the three exposure maps and infection distribution bar charts might suggest a reduction in the potential infection area at maximum velocity, it is essential to note that higher airspeed causes droplets to travel greater distances. In reality, this does not lead to a decrease in potential infection risk instead.

Based on this comparative analysis, it is evident that at an inlet air velocity of 0.77 m/s, the potential infection area is comparatively narrower than under the other two velocity conditions. This implies that a much higher air velocity does not necessarily confer an advantage. Therefore, we recommend maintaining an optimal air inlet velocity of approximately 0.77 m/s.

In addition, it can be found that in the previous method evaluation, the virus amount of escape was also included actually, which may result the actual infection probability of the human to be lower than the actual infection probability. Meanwhile, most studies lack of analysis of exposure risk. However, the new method—*PCR* has the potential to shift the focus from a global perspective to a local one. This shift aims to avoid the influence of escape viruses and trap viruses by the cabin wall in the assessment process, and fill the gap in the supplementary exposure risk section. Additionally, the method integrates the modified Wells-Riley model to redistribute the collective infection risk, enabling a reasonable evaluation of the relative infection risk for a specific individual rather than a collective assessment for a group of people.

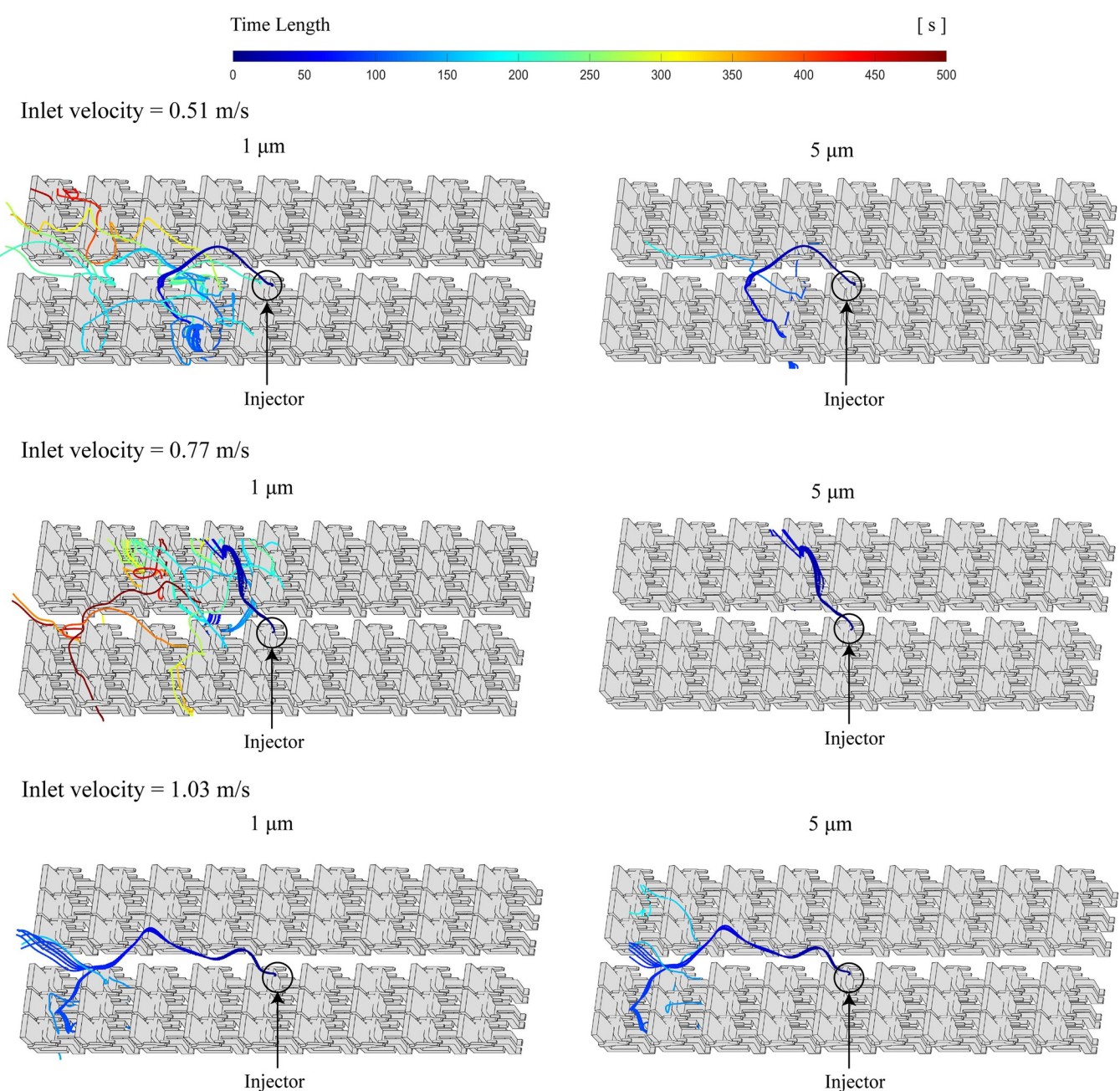

**Fig 11. The droplets trajectories in three different cases of inlet velocity.**

## 4. Conclusion

In an effort to support aircraft designers in optimizing cabin components such as ventilation systems and seat arrangements and to assist health authorities in developing more effective pandemic prevention strategies, particularly in cases of outbreaks such as COVID-19 and Influenza, this research utilizes computational fluid dynamics (CFD) techniques. The primary focus is to investigate the potential risks of airborne contamination within individual breathing zones under different inlet air velocities. To quantify these risks, we introduce and apply the *Personal Contamination Ratio* (*PCR*) method. Our findings reveal differences in potential

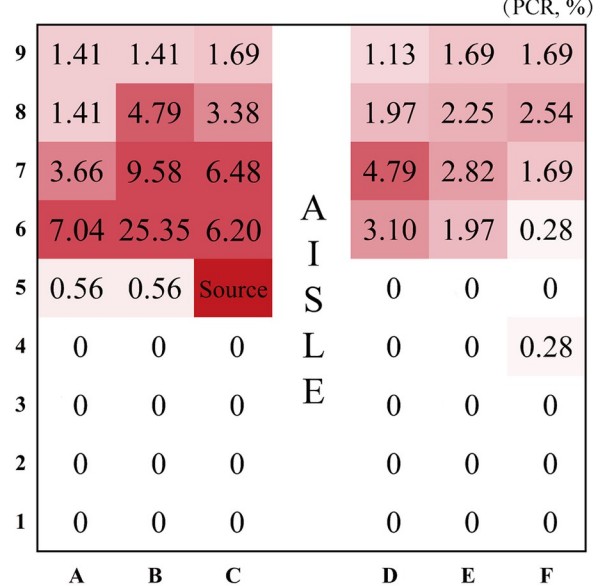

Inlet Velocity = 0.51 m/s

Inlet Velocity = 0.77 m/s

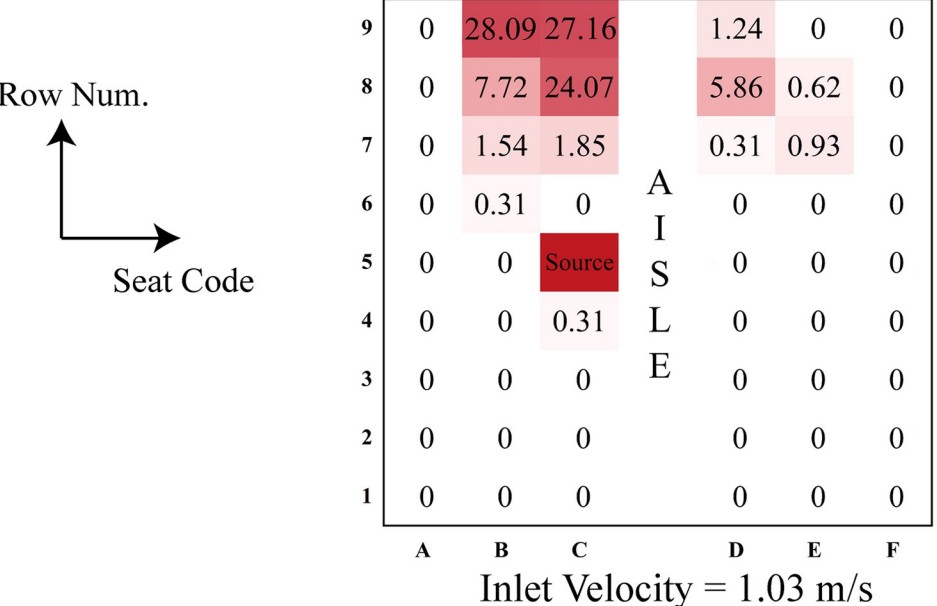

Inlet Velocity = 1.03 m/s

**Fig 12. Variation maps in exposure risk within the passenger breathing zone across different inlet velocities.**

infection risks in different areas of the cabin under different velocity conditions. Based on the results of CFD experiments, we draw the following conclusions:

1. The direction of airflow within the aircraft cabin is closely linked to the inlet velocity, a critical parameter in cabin ventilation systems. Our investigation reveals that air supplied through the ventilation systems converges near the lower section of the aisle before ascending, regardless of the inlet velocity. This airflow pattern is significantly influenced by

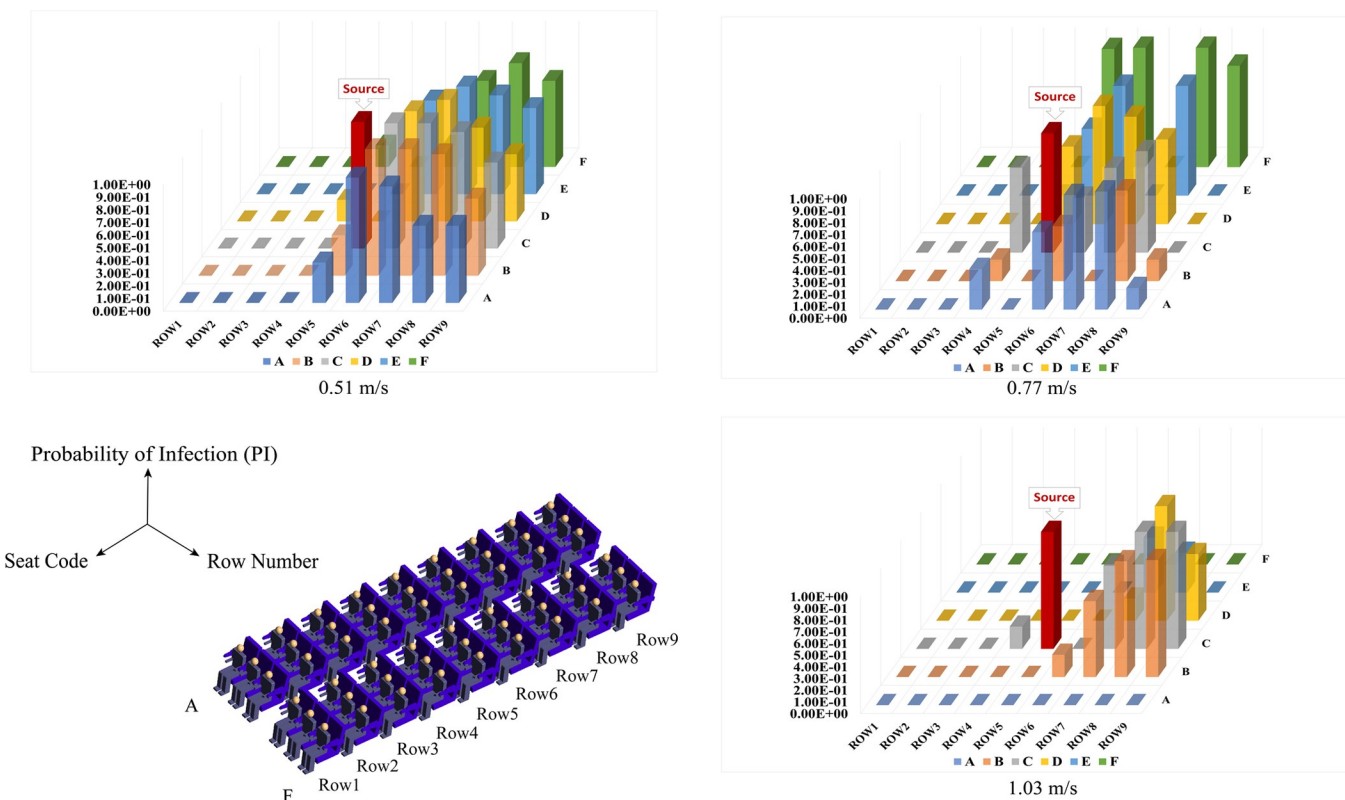

**Fig 13. Three-dimensional bar chart based on the relative probability of infection distribution.**

thermal plumes generated by human occupants and their respiratory processes. Notably, as the inlet velocity increases, the rearward flow of air becomes more pronounced. These observations underscore the complex interplay between inlet velocity, thermal plumes, and the resultant airflow patterns within the cabin environment.

2. The trajectories of small-sized droplets exhibit high sensitivity to variations in inlet air velocity. As the inlet air supply incrementally increases from its minimum threshold, these droplets demonstrate a tendency to converge, resulting in a relative reduction of potential infection areas. In contrast, larger droplets display a slower response to changes in inlet air velocity. Their trajectories undergo significant alterations only in response to substantial increases in inlet air velocity.

3. Optimizing the inlet air velocity is crucial for effective infection control. Experimental results demonstrate that the minimum inlet velocity leads to the largest potential infection area. Conversely, maximum velocity reduces the potential infection area in the transverse direction but significantly increases it longitudinally. Notably, the number of potential infections within the cabin at the highest inlet air velocity decreases by 51.8% compared to the basic inlet air velocity, and the total exposure risk rate is reduced by 26%. Furthermore, the number of rows with infected passengers decreases to 4 rows under maximum inlet air velocity. The probability of infection for more than 50% of individuals is reduced by 65.2% at the highest inlet air velocity and by 21.74% at moderate inlet air velocity compared to the basic inlet air velocity. Additionally, the total number of individuals at risk of infection is nearly halved at the highest inlet air velocity. Considering passenger comfort alongside

infection control, operating at a moderate velocity emerges as the optimal strategy. This balanced approach effectively mitigates infection risk while maintaining acceptable comfort levels for passengers. In this study, only one pathogen-releasing CSP, seated in position 5C, was considered.

In this study, we introduce a new approach, the *Personal Contamination Ratio* (*PCR*) method, to quantitatively assess individual infection risks. This method offers a more detailed assessment of specific personal risk areas, distinguishing it from previous approaches. The outcomes of this research can serve as a valuable reference for optimizing cabin air inlet velocities and implementing protective measures to safeguard potentially infected individuals. In this study, we did not explicitly account for the effects of humidity on droplet transport within the aircraft cabin environment. Our decision was informed by previous research indicating that droplets tend to evaporate rapidly in the extremely dry conditions of an aircraft cabin, thereby minimizing the impact of ambient humidity on droplet dynamics. This simplification was necessary to focus on the primary objective of our research: the development and validation of the Personal Contamination Ratio (*PCR*) method. While this approach allowed us to highlight the efficacy of the *PCR* method, we recognize that excluding humidity and evaporation effects may limit the generalizability of our findings to other environments. Future studies should consider more factors to provide a more comprehensive understanding of droplet behavior under varying humidity conditions.

## Supporting information

**S1 Data. Docx for Fig 2.**
(DOCX)

**S2 Data. Docx for Fig 5.**
(DOCX)

**S3 Data. Docx for Fig 13.**
(DOCX)

## Author Contributions

**Conceptualization:** Renquan Tu, Yidan Shang.

**Data curation:** Renquan Tu.

**Funding acquisition:** Yidan Shang, Fajiang He, Jiyuan Tu.

**Investigation:** Renquan Tu, Yidan Shang, Xueren Li, Fajiang He.

**Methodology:** Renquan Tu, Yidan Shang, Xueren Li.

**Project administration:** Yidan Shang.

**Software:** Renquan Tu.

**Supervision:** Yidan Shang.

**Validation:** Renquan Tu, Yidan Shang, Xueren Li, Fajiang He.

**Visualization:** Renquan Tu.

**Writing – original draft:** Renquan Tu.

**Writing – review & editing:** Yidan Shang, Xueren Li, Jiyuan Tu.

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
