## [Decision Letter · Decision Letter 0]

5 Mar 2024

PONE-D-24-00989Optimizing cabin air inlet velocities and personal risk assessment: introducing the Personal Contamination Ratio (PCR) method for enhanced aircraft cabin infection risk evaluationPLOS ONE

Dear Dr. Shang,

Thank you for submitting your manuscript to PLOS ONE. After careful consideration, we feel that it has merit but does not fully meet PLOS ONE’s publication criteria as it currently stands. Therefore, we invite you to submit a revised version of the manuscript that addresses the points raised during the review process. Please submit your revised manuscript by Apr 19 2024 11:59PM. If you will need more time than this to complete your revisions, please reply to this message or contact the journal office at plosone@plos.org. Please include the following items when submitting your revised manuscript:A rebuttal letter that responds to each point raised by the academic editor and reviewer(s). You should upload this letter as a separate file labeled 'Response to Reviewers'.A marked-up copy of your manuscript that highlights changes made to the original version. You should upload this as a separate file labeled 'Revised Manuscript with Track Changes'.An unmarked version of your revised paper without tracked changes. You should upload this as a separate file labeled 'Manuscript'.

We look forward to receiving your revised manuscript.

Kind regards,

Krit Pongpirul, MD, MPH, PhD.

Academic Editor

PLOS ONE

Journal Requirements:

"This research was funded by the National Natural Science Foundation of China (Grant No. 82370101) and the Program for Professor of Special Appointment (Eastern Scholar) at Shanghai Institutions of Higher Learning (Project ID: 0920000016)."

"This research was funded by the National Natural Science Foundation of China (Grant No. 82370101) and the Program for Professor of Special Appointment (Eastern Scholar) at Shanghai Institutions of Higher Learning (Project ID: 0920000016). The funders had no role in study design, data collection and analysis, the decision to publish, or the preparation of the manuscript."

Reviewers' comments:

Reviewer's Responses to Questions

**Comments to the Author**

1. Is the manuscript technically sound, and do the data support the conclusions?

Reviewer #1: Yes

Reviewer #2: No

2. Has the statistical analysis been performed appropriately and rigorously? 

Reviewer #1: Yes

Reviewer #2: N/A

3. Have the authors made all data underlying the findings in their manuscript fully available?

Reviewer #1: Yes

Reviewer #2: Yes

4. Is the manuscript presented in an intelligible fashion and written in standard English?

Reviewer #1: No

Reviewer #2: Yes

5. Review Comments to the Author

Reviewer #1: This paper reports the results of a series of numerical analysis of airborne transmission in a enclosed space, assuming an aircraft cabin. The results of the numerical analysis are compared with experimental results for a mock-up cabin model, and a certain degree of validation of the prediction accuracy of numerical method is also provided. The infection risk model, although simple, is a model that has been applied widely, and a certain level of prediction accuracy is ensured.

The research theme and the numerical analysis method are not claimed to be novel, since they are based on previous studies, but they contain some novelty compared to previous studies, such as the use of a new index for evaluating exposure concentrations, PCR.

Overall, this manuscript is judged to have few shortcomings and to be well-considered/well-organized, and this reviewer would recommend this should be accepted.

Reviewer #2: The present study performs numerical simulations to analyse droplet dynamics in a Airbus A320 model, evaluating the effect of changing air supply rate on passengers’ exposure to droplets. The topic is interesting and worth of investigation; however, there are issues associated with the methodology and the CFD simulations, making the proposed results not reliable and the paper unsuitable for publication. Listed below are the main criticalities of the paper.

- The Authors present the analysis of the breathing zone as a strong point, but the dynamics of breathing and particle inhalation are completely neglected. These aspects considerably alters the distribution of particles in the breathing zone, making the results obtained unreliable.

- English language needs improvement. At some points, the sentence construction is convoluted; contracted forms are also present, which are not suitable for a scientific paper.

- The description of the simulated scenario is confusing. It is stated that 6 people have been simulated (line 152), but in Figure 1 more passengers are depicted. In addition, authors should demonstrate that the simulated domain (restricted to a section of the whole cabin) is representative of the problem under investigation.

- Authors should specify the software employed to carry out CFD simulations.

- It is not sufficient to state that “Building upon existing research findings, our simulations employed a mesh grid totaling 6.51 million polyhedral cells within the cross-section of the three-row cabin” (lines 152-154). Unless the authors have carried out previous studies on the same cabin model, a grid sensitivity analysis must be performed.

- A complete description of the boundary conditions should be provided. The periodic boundary condition set for the “periodic faces” needs to be discussed further.

- The governing equations presented in section 2.4 (Equation 1 and 2) are for a laminar, incompressible and unsteady flow. These are not the governing equations of the problem under investigation. The URANS equations must be provided.

In addition, the Boussinesq approximation seems to be used to model the effects of buoyancy (by considering the density constant in the transient and convective terms), but then in the gravitational term its dependence on temperature is not considered.

- The drag force (equation 3) should be evaluated as a function of the droplet Reynolds number.

- The scenario analyzed for particle emission is not realistic at all. It should also be described in more detail, not simply reporting the total number of particles emitted.

The diameters (1 and 5 µm) are not representative of a real scenario, nor are the velocity (fixed at 1 m/s) and the direction of release.

- How would the model proposed by the Authors improve the Wells-Riley model, by changing the perspective from global to local (section 2.5)?

In the Referee’s opinion, the proposed model is not reliable to provide quantitative information about the risk of infection; rather, it gives information about the relative weight between different zones. For this purpose, it would have been sufficient to show the concentration of particles in different zones.

- What is the error between PIV measurements and CFD results? The agreement seems to be very poor, especially for lines A and F (Figure 4). Such disagreement raises questions regarding the turbulence model, the boundary conditions and the grid sensitivity analysis.

In addition, the section of the experimental-numerical comparison should be highlighted in the computational domain and the scenario considered for the validation should be described.

- I find the representation in Figure 5 of little use, with the experimental vectors barely visible. The legend is also missing (as in the later images). It would be more useful to represent the entire measured velocity field.

In general, the representation with vectors is unreadable and does not allow to visualize the presence of recirculation zones; in this sense, streamlines would be more suitable.

- A picture depicting particle spatial distribution should be provided, commented with reference to the simulated velocity fields.

- A strange effect is present in the velocity fields of the simulated scenarios whereby the air jet is strongly drawn back to the wall, despite being released at a certain angle toward the inside of the cabin. This effect is not present in the validation scenario and should be explained by the authors.

- How is it possible for air to be completely carried behind (resulting in infection only for those passengers sitting behind the infected person)? The explanation provided by the authors (lines 379-383) is not convincing and the effect of the boundary conditions set at the "periodic faces" should be investigated.

- Representing the possibility of infection for the source (Figure 12) makes no sense. In addition, there is probably a typo in the legend (the highest probability is equal to 0.25%, which is very low).

6. PLOS authors have the option to publish the peer review history of their article (what does this mean?). If published, this will include your full peer review and any attached files.

Reviewer #1: No

Reviewer #2: No

---

## [Author Response · Author response to Decision Letter 0]

13 May 2024

Response to Reviewer #1's Comments

We're truly grateful for the reviewer's supportive feedback, which means a lot to us.

Response to Reviewer #2's Comments

Comments:

1. The Authors present the analysis of the breathing zone as a strong point, but the dynamics of breathing and particle inhalation are completely neglected. These aspects considerably alters the distribution of particles in the breathing zone, making the results obtained unreliable.

We thank the reviewers for raising the insightful comments. The authors would like to clarify the major objective of this study is to propose a novel holistic infection risk assessment framework by incorporating the concept of the breathing zone and the Personal Contamination Ratio (PCR) into the original Wells-Riley model. Based on the detailed spatial-temporal information of the tracked droplets from the CFD simulations, it allowed us to obtain a more reasonable infection risk quantification considering the particle distribution.

We acknowledged the significance of considering dynamics related to breathing and particle inhalation, which was believed to enhance the overall robustness of the assessment. While those aspects were believed to be beneficial, including them into the simulation was expected to significantly improve the overall computational cost. Considering the complex transient nature of the breathing patterns (boundary condition variations), and multi-scale (from cabin environment, m3, to human nasal, mm3) and multi-coupling challenges, further including those setups within a densely-occupied cabin space was expected to exponentially increase the simulation difficulty and time. Moreover, many existing studies overlook the influence of the breathing pattern, as exemplified by the following instances: 

[1] Kong B, Zou Y, Cheng M, Shi H, Jiang YJAS. Droplets transmission mechanism in a commercial wide-body aircraft cabin. 2022;12(10):4889.

[2] Wang F, Zhang TT, You R, Chen Q. Evaluation of infection probability of Covid-19 in different types of airliner cabins. Build Environ. 2023;234:110159.

[3] You R, Lin CH, Wei D, Chen QJIA. Evaluating the commercial airliner cabin environment with different air distribution systems. 2019;29(5):840-53.

[4] Choi E-S, Yook S-J, Kim M, Park DJT. Study on the Ventilation Method to Maintain the PM10 Concentration in a Subway Cabin below 35 μg/m3. 2022;10(10):560.

[5] Zhang M, Yu N, Zhang Y, Zhang X, Cui YJP. Numerical simulation of the novel coronavirus spread in commercial aircraft cabin. 2021;9(9):1601.

2. English language needs improvement. At some points, the sentence construction is convoluted; contracted forms are also present, which are not suitable for a scientific paper.

Thanks for raising this issue which we agree with. We checked the grammar and wording in the manuscript and revised them accordingly, as follows:

 Page 2, lines 39-40. ‘Air cabins play an important role in spreading infectious diseases (8-11).’

 Page 3, lines 63-64. ‘However, Yan, Li (18)'s investigations predominantly focused on droplets of a representative size, which may not be widely applicable to all scenarios.’

 Page 10, lines 281-283. ‘Compared to previous methods used for predicting droplet spread, this study introduces a new exposure risk quantification method which named PCR (Personal Contamination Ratio).’

 Page 12, lines 347-348. ‘Consequently, this research begins with visualizing cabin airflow characteristics to better understanding of the particle distribution.’

 Page 13, lines 374-375. ‘The distribution of particle in the cabin is influenced not only by inlet air velocity but also by the direction of airflow.’

 Page 17, lines 488-489. ‘Their trajectories are altered significantly in response to a substantial increase in inlet air velocity.’

3. The description of the simulated scenario is confusing. It is stated that 6 people have been simulated (line 152), but in Figure 1 more passengers are depicted. In addition, authors should demonstrate that the simulated domain (restricted to a section of the whole cabin) is representative of the problem under investigation.

Thanks for the reviewer’s comment. The authors would like to clarify that in this study, a total of 18 passengers were modelled within an aircraft cabin. 6 passengers were constructed in a single row with one aisle to mimic the real cabin layout. Based on the review’s comment, the authors found that the previous description of the model can be quite confusing for the readers and such description has now been revised based on the reviewer’s comment (please see page6, lines 154-155). 

‘As shown in Figure 1, passengers and seats are arranged on two sides of the cabin. There are three rows, each accommodating six persons, making a total number of 18 passengers.’

As for the second concern, the authors acknowledged that investigating the in-cabin aerosol transmission scenario with a full-scale cabin model would be perfect as it provides a comprehensive picture of the particle transport and distribution characteristics. However, the authors would like to point out that restoring such phenomena can be really time-consuming (e.g. solving the large-scale, multi-phase simulation). In this study, the authors adopted the aforementioned layout as recent studies demonstrate that severe transmission generally occurs longitudinally, within the 3 rows [1]. Most existing studies are based on such outcomes to model the in-cabin transmission scenarios in recent years [2-5]. The authors acknowledge that the current cabin model would not be perfect, while based on the existing findings and mainstream studies, such a computational model was expected to be reasonable and representative. Additionally, we have added some text in page 5, lines 137-139. Thanks for the reviewer’s comment.

‘Building upon previous literature (18, 26, 27, 34, 35), to optimize computational resources, we have chosen to focus our analysis on a representative subset of the cabin section with three rows.’

[1] Silcott, David, et al. "TRANSCOM/AMC commercial aircraft cabin aerosol dispersion tests." (2020).

[2] Zhao, Yingjie, et al. "Numerical simulation study on air quality in aircraft cabins." Journal of Environmental Sciences 56 (2017): 52-61.

[3] Zee, Malia, et al. “Computational fluid dynamics modeling of cough transport in an aircraft cabin.” Scientific reports 11.1 (2021): 23329.

[4] Kong, Benben, et al. “Droplets transmission mechanism in a commercial wide-body aircraft cabin.” Applied Sciences 12.10 (2022): 4889.

[5] Yan, Yihuan, et al. "Evaluation of cough-jet effects on the transport characteristics of respiratory-induced contaminants in airline passengers’ local environments." Building and environment 183 (2020): 107206.

4. Authors should specify the software employed to carry out CFD simulations.

We appreciate the reviewer for reminding this issue. We have added specific details regarding the software used for the CFD simulations on page 8, lines 222-223.

‘This study utilized the commercial computational fluid dynamics (CFD) software ANSYS Fluent 2021 R1 for conducting all numerical calculations.’

5. It is not sufficient to state that “Building upon existing research findings, our simulations employed a mesh grid totaling 6.51 million polyhedral cells within the cross-section of the three-row cabin” (lines 152-154). Unless the authors have carried out previous studies on the same cabin model, a grid sensitivity analysis must be performed.

We appreciate the reviewer for bringing up this issue. We acknowledge the importance of conducting mesh sensitivity analysis for mesh generation, and we have provided the information about the mesh sensitivity analysis below. The horizontal axis of the table represents velocity, while the vertical axis represents the length of the validation line. Through the comparison from this picture, there is no considerable deviation in the velocity field noticed after increasing the number of mesh elements from 6.51 million to 7.84 million. Thus, we chose to use a mesh setup with 6.51 million elements for the simulations. Additionally, in the manuscript on page 6, lines 155-162 and 179-180, we have included a description of the mesh independence test as follow:

‘To achieve mesh independence, this study experimented with five sets of mesh configurations, utilizing total mesh elements of 2.60 million, 3.92 million, 5.20 million, 6.51 million, 7.84 million, respectively. Significant discrepancies in velocity were observed among the groups with grid numbers of 2.60, 3.92, and 5.20 million, while an increase in mesh elements from 6.51 million to 7.84 million revealed negligible deviation in the velocity field, as illustrated in Figure 3. This indicates that a mesh of 6.51 million elements is suitable for simulating airflow within the computer cabin. Therefore, in our simulations employed a mesh grid totaling 6.51 million polyhedral cells within the cross-section of the three-row cabin, achieving a maximum skewness of 0.81.’

Please see picture in the Response to Reviewers.docx

6. A complete description of the boundary conditions should be provided. The periodic boundary condition set for the “periodic faces” needs to be discussed further.

We agree with the constructive suggestion. The detailed description of the periodic boundary condition is added to page 7, line 200.

‘…we have set the floor, seats, walls, and ceiling as no-slip surfaces. Additionally, we employed translation as the periodic face type.’

7. The governing equations presented in section 2.4 (Equation 1 and 2) are for a laminar, incompressible and unsteady flow. These are not the governing equations of the problem under investigation. The URANS equations must be provided.

In addition, the Boussinesq approximation seems to be used to model the effects of buoyancy (by considering the density constant in the transient and convective terms), but then in the gravitational term its dependence on temperature is not considered.

Thanks for raising this issue. We have revised the formula, with specific modifications outlined on page 8, lines 226-233, as follows:

Please see detailes in the Response to Reviewers.docx

Additionally, we sincerely apologize for any confusion caused to readers due to our oversight. The misleading descriptions in the manuscript have now been corrected, as evidenced on page 8, lines 223-224 as follows:

‘The Navier-Stokes(N-S) equations with the Boussinesq approximation were used to simulate airflow field in Eulerian method’

8. The drag force (equation 3) should be evaluated as a function of the droplet Reynolds number.

Thanks for your reminder. The Reynolds number is already included in the formula. To enhance the clarity and conciseness of the formula expression, the first term on the right-hand side of the equation is separated and presented individually. 

Please see equation in the Response to Reviewers.docx

9. The scenario analyzed for particle emission is not realistic at all. It should also be described in more detail, not simply reporting the total number of particles emitted.

The diameters (1 and 5 µm) are not representative of a real scenario, nor are the velocity (fixed at 1 m/s) and the direction of release.

Thank you for bringing up this question. We would like to clarify that the details of the particle information can be found in Figure 1, specifying both the location and angle of incidence. Additionally, we have provided further details regarding the particle emission location on pages 5, lines 146-151. The specifics are as follow. 

In consideration of the velocity and direction of release, we carefully referred to peer-reviewed literature and set up the specifics based on a scenario involving a stationary seated individual in a steady state, as outlined in reference [3-4]. We believe this adequately explains the information regarding the particle inlet. 

‘Following the recommendations of Kong, Zou (26), we have represented the CSP model's particle injection, located on the face, as a simplified circular inlet with a diameter of 1.24 cm. In accordance with findings from Haselton and Sperandio (36) and Kuga, Wargocki (28), the exhalation zone is identified to span from 27° and 33°. This study set a central value of 30° as the particle injection angle.’

As for the particle diameter, we have provided references as follow to support the significance of investigating diameters of 1 and 5 µm. In particular, the reference - [5] illustrates that the particle diameter of human respiratory particles is almost below 8 µm:

[1] Thomas, Richard James. "Particle size and pathogenicity in the respiratory tract." Virulence 4.8 (2013): 847-858.

[2] Shang, Y. D., K. Inthavong, and J. Y. Tu. "Detailed micro-particle deposition patterns in the human nasal cavity influenced by the breathing zone." Computers & Fluids 114 (2015): 141-150.

[3] Kuga K, Wargocki P, Ito KJIa. Breathing zone and exhaled air re‐inhalation rate under transient conditions assessed with a computer‐simulated person. 2022;32(2):e13003.

[4] Zhang M, Yu N, Zhang Y, Zhang X, Cui YJP. Numerical simulation of the novel coronavirus spread in commercial aircraft cabin. 2021;9(9):1601.

[5] Zhang, Hualing, et al. "Documentary research of human respiratory droplet characteristics." Procedia engineering 121 (2015): 1365-1374.

10. How would the model proposed by the Authors improve the Wells-Riley model, by changing the perspective from global to local (section 2.5)?

In the Referee’s opinion, the proposed model is not reliable to provide quantitative information about the risk of infection; rather, it gives information about the relative weight between different zones. For this purpose, it would have been sufficient to show the concentration of particles in different zones.

Thank you for your insightful question and feedback. The approach introduced in our study offers a significant departure from the global perspective typically employed in prior research. By modifying the methodology for calculating exposure risk (as highlight in the formula) and integrating the PCR factor into the infection risk calculation, we shift the focus from a global outlook to an individualized perspective. This shift enables us to accurately quantify the localised infection risk of each seated passenger rather than analysing the global infection risk, thereby enhancing the reliability of the quantitative information provided. 

As for the second concern, we would like to clarify that the method employed in our study, the Eula-Laglang method, is specially tailored to analyze particle trajectory paths. Our primary objective is to comprehensively investigate particle trajectories from infection source to each individual's breathing zone. Within this framework, we have quantified particle numbers within each breathing zone to assess infection and exposure risks across different areas. Besides, we have revised the calculation formula for Possibility of Infection to ensure a more accurate characterization of each individual’s PI value.

The way to improve the Wells-Riley model in this study:

As for the improvement to the Wells-Riley model, we specifically adjust the calculation of the personal contamination ratio by shifting from considering the total particle amount in a zone to evaluating the particle concentration within each individual's breathing zone.

Please see equations in the Response to Reviewers.docx

Among them, N_vs (V) indicates the cumulative number of droplets in a single breathing zone, and N_vr (V) represents the cumulative number of droplets particles in all breathing zones. C represents the droplets concentration in the droplets discharged at the moment of discharge, and d_i is the original diameter of the i th expelled droplet.

Please see equations in the Response to Reviewers.docx

Among them, θ denotes the ratio coefficient of 〖HID〗_50 (median infective dose in humans) to 〖TCID〗_50, specifically we utilize the influenza data for estimation. N_particles represents the number of the particles, p signifies the pulmonary ventilation rate, and t represents the duration of the flight.

11. What is the error between PIV measurements and CFD results? The agreement seems to be very poor, especially for lines A and F (Figure 4)

---

## [Decision Letter · Decision Letter 1]

19 Jul 2024

PONE-D-24-00989R1Optimizing cabin air inlet velocities and personal risk assessment: introducing the Personal Contamination Ratio (PCR) method for enhanced aircraft cabin infection risk evaluationPLOS ONE

Dear Dr. Shang,

Thank you for submitting your manuscript to PLOS ONE. After careful consideration, we feel that it has merit but does not fully meet PLOS ONE’s publication criteria as it currently stands. Therefore, we invite you to submit a revised version of the manuscript that addresses the points raised during the review process.

We look forward to receiving your revised manuscript.

Kind regards,

Krit Pongpirul, MD, MPH, PhD.

Academic Editor

PLOS ONE

Additional Editor Comments:

Please carefully address the additional comments from the reviewers.

Reviewers' comments:

Reviewer's Responses to Questions

**Comments to the Author**

1. If the authors have adequately addressed your comments raised in a previous round of review and you feel that this manuscript is now acceptable for publication, you may indicate that here to bypass the “Comments to the Author” section, enter your conflict of interest statement in the “Confidential to Editor” section, and submit your "Accept" recommendation.

Reviewer #3: All comments have been addressed

Reviewer #4: (No Response)

2. Is the manuscript technically sound, and do the data support the conclusions?

Reviewer #3: Yes

Reviewer #4: Partly

3. Has the statistical analysis been performed appropriately and rigorously? 

Reviewer #3: Yes

Reviewer #4: No

4. Have the authors made all data underlying the findings in their manuscript fully available?

Reviewer #3: Yes

Reviewer #4: Yes

5. Is the manuscript presented in an intelligible fashion and written in standard English?

Reviewer #3: Yes

Reviewer #4: Yes

6. Review Comments to the Author

Reviewer #3: The authors provide references to sources providing information about the conditions representing the tested case.

The authors provided drawings and data explaining the adopted methods and results to demonstrate the usefulness of the PCR method.

This study contributes important information on droplet transport dynamics and infection risk in aircraft cabins, highlighting the importance of optimizing air delivery rates. PCR can significantly contribute to better design of ventilation systems and public health strategies.

Reviewer #4: The manuscript entitled “Optimizing cabin air inlet velocities and personal risk assessment: introducing the Personal Contamination Ratio (PCR) method for enhanced aircraft cabin infection risk evaluation” is interesting. However, there are several major concerns that authors shall address.

1. Abstract: Measurable findings are required in the abstract session. Conclusion Is not clearly highlighted as in current form.

2. Introduction- suggest adding on other airborne infection statistics and its description, instead of COVID-19 only. These inclusions could show the importance of present study to be adopted in future.

3. Description of droplets shall be included, i.e., material, density, viscosity, other physical properties.

4. Is humidity being considered in this study? If no, please justify thoroughly. As far as reviewer concern, humidity could significantly affect the droplets transportation characteristics.

5. Authors shall justify why SIMPLE scheme for pressure velocity coupling and second order upwind scheme are chosen. Else, authors might need to find reference (similar study that investigate the effect of droplets dispersion in indoor) to support. Example: https://doi.org/10.1016/j.enbuild.2023.113439

6. Line 340- satisfactory agreement could be subjective. What is the relative error? This information is very crucial to support the reliability of result.

7. PLOS authors have the option to publish the peer review history of their article (what does this mean?). If published, this will include your full peer review and any attached files.

Reviewer #3: **Yes: **Konrad Gumowski

Reviewer #4: No

---

## [Author Response · Author response to Decision Letter 1]

2 Aug 2024

Ref: PONE-D-24-00989

Title: Optimizing cabin air inlet velocities and personal risk assessment: introducing the Personal Contamination Ratio (PCR) method for enhanced aircraft cabin infection risk evaluation

Dear Editor and Reviewers,

We greatly appreciate your valuable comments and suggestions which have helped enhance our manuscript. Based on your feedback, we have made improvements to our work. We have addressed each of your points in detail, with changes highlighted in red in the revised manuscript and responses marked in blue below.

Reviewer #3's Comments

The authors provide references to sources providing information about the conditions representing the tested case.

The authors provided drawings and data explaining the adopted methods and results to demonstrate the usefulness of the PCR method.

This study contributes important information on droplet transport dynamics and infection risk in aircraft cabins, highlighting the importance of optimizing air delivery rates. PCR can significantly contribute to better design of ventilation systems and public health strategies.

We are truly grateful for the reviewer's supportive feedback, which means a lot to us. We appreciate the time and effort invested in reviewing our manuscript, and we are pleased that the revisions have met with approval. Thank you for recognizing the value of our research and for your encouragement.

 

Reviewer #4's Comments

The authors would like to express their sincere gratitude to the reviewer for providing thoughtful and valuable feedback on this manuscript. The improvements made to the manuscript were greatly influenced by the reviewer's insightful comments.

The manuscript entitled “Optimizing cabin air inlet velocities and personal risk assessment: introducing the Personal Contamination Ratio (PCR) method for enhanced aircraft cabin infection risk evaluation” is interesting. However, there are several major concerns that authors shall address.

1.Abstract: Measurable findings are required in the abstract session. Conclusion Is not clearly highlighted as in current form.

Response: We appreciate the reviewer for reminding this issue. Based on the reviewer’s comment, we have revised the content of the abstract section as follows:

‘Recurrent epidemics of respiratory infections have drawn attention from the academic community and the general public in recent years. Aircraft plays a pivotal role in facilitating the cross-regional transmission of pathogens. In this study, we initially utilized an Airbus A320 model for computational fluid dynamics (CFD) simulations, subsequently validating the model's efficacy in characterizing cabin airflow patterns through comparison with empirical data. Building upon this validated framework, we investigate the transport dynamics of droplets of varying sizes under three air supply velocities. The Euler-Lagrangian method is employed to meticulously track key parameters associated with droplet transport. This study integrates acquired data into a novel PCR (Personal Contamination Rate) equation to assess individual contamination rates. Numerical simulations demonstrate that increasing air supply velocity leads to enhanced stability in the movement of larger particles compared to smaller ones. Results show that the number of potential infections in the cabin decreases by 51.8 % at the highest air supply velocity compared to the base air supply velocity, and the total exposure risk rate reduced by 26.4 %. Thus, optimizing air supply velocity within a specific range effectively reduces the potential infection area. In contrast to previous research, this study provides a more comprehensive analysis of droplet movement dynamics across various particle sizes. We introduce an improved method for calculating the breathing zone, thereby enhancing droplet counting accuracy. It is hoped that these findings can provide valuable insights for enhancing non-pharmacological public health interventions and improving cabin ventilation system design.’

Please see Page 1, lines 11-29 in the revised manuscript.

Additionally, we have also revised the content of the conclusion section as follows:

‘(1) The direction of airflow within the aircraft cabin is closely linked to the inlet velocity, a critical parameter in cabin ventilation systems. Our investigation reveals that air supplied through the ventilation systems converges near the lower section of the aisle before ascending, regardless of the inlet velocity. This airflow pattern is significantly influenced by thermal plumes generated by human occupants and their respiratory processes. Notably, as the inlet velocity increases, the rearward flow of air becomes more pronounced. These observations underscore the complex interplay between inlet velocity, thermal plumes, and the resultant airflow patterns within the cabin environment.

(2) The trajectories of small-sized droplets exhibit high sensitivity to variations in inlet air velocity. As the inlet air supply incrementally increases from its minimum threshold, these droplets demonstrate a tendency to converge, resulting in a relative reduction of potential infection areas. In contrast, larger droplets display a slower response to changes in inlet air velocity. Their trajectories undergo significant alterations only in response to substantial increases in inlet air velocity.

(3) Optimizing the inlet air velocity is crucial for effective infection control. Experimental results demonstrate that the minimum inlet velocity leads to the largest potential infection area. Conversely, maximum velocity reduces the potential infection area in the transverse direction but significantly increases it longitudinally. Notably, the number of potential infections within the cabin at the highest inlet air velocity decreases by 51.8 % compared to the basic inlet air velocity, and the total exposure risk rate is reduced by 26 %. Furthermore, the number of rows with infected passengers decreases to 4 rows under maximum inlet air velocity. The probability of infection for more than 50% of individuals is reduced by 65.2 % at the highest inlet air velocity and by 21.74 % at moderate inlet air velocity compared to the basic inlet air velocity. Additionally, the total number of individuals at risk of infection is nearly halved at the highest inlet air velocity. Considering passenger comfort alongside infection control, operating at a moderate velocity emerges as the optimal strategy. This balanced approach effectively mitigates infection risk while maintaining acceptable comfort levels for passengers. In this study, only one pathogen-releasing CSP, seated in position 5C, was considered.’

Please see Page 17, lines 488-502; Page 18, lines 503-515 in the revised manuscript.

2. Introduction- suggest adding on other airborne infection statistics and its description, instead of COVID-19 only. These inclusions could show the importance of present study to be adopted in future.

Response: Thanks for bring up this issue. We have searched the statistical data on various respiratory diseases and have provided a specific example for context. The details of the revision are as follows:

‘Airborne infectious diseases pose a significant threat to global public health, with the potential to spread rapidly and overwhelm healthcare systems during outbreaks. These respiratory-borne illnesses can lead to widespread infections, affecting millions of people worldwide each year. The transmission of such diseases has become a focal point of research, particularly in light of recent global health crises. Among the various airborne infectious diseases, some stand out due to their impact and prevalence. For instance, Respiratory Syncytial Virus (RSV) spreads extensively during certain periods each year, with a median duration of 4.6 months, affecting millions globally [1]. More recently, the COVID-19 pandemic has had an unprecedented global impact, resulting in 775.69 million confirmed cases and 6.95 million deaths. This pandemic has significantly raised awareness about the transmission of respiratory diseases and led to extensive research and vaccination campaigns. The emergence of influenza during the later stages of the COVID-19 pandemic further highlights the threat posed by respiratory diseases. These outbreaks have intensified the focus on transmission studies, revealing the potential for rapid spread and severe strain on healthcare resources. In the current globalization period, air travel in particular has become an essential way of linking people worldwide. In this mode of transportation, air cabin plays an important role in spreading infectious diseases.’

Please see Page 2, lines 32-47 in the revised manuscript.

Additionally, to support our findings, we have included relevant literature reference, which is listed below:

[1] Agca H, Akalin H, Saglik I, Hacimustafaoglu M, Celebi S, Ener B. Changing epidemiology of influenza and other respiratory viruses in the first year of COVID-19 pandemic. Journal of Infection and Public Health. 2021;14(9):1186-90.

3. Description of droplets shall be included, i.e., material, density, viscosity, other physical properties.

Response: Thank you for bringing this important issue to our attention. We agree that it is crucial for accurate simulation. Based on equivalent aerodynamic principles, this approach simplifies and predicts the motion, settling, and spreading behavior of real particles or droplets by comparing them to idealized spherical particles with the same aerodynamic properties. In our study, we defined droplets as inert particles and modeled them with a density of 1000 kg/m³ to reflect their water-liquid composition. Based on the reviewer’s comment, we have added a detailed description of the droplets as follows:

‘To simulate the droplets based on equivalent aerodynamic principles, we defined them as inert particles, allowing us to focus on their aerodynamic behavior and physical characteristics without involving complex chemical reactions or biodegradation processes. Based on previous research, the droplets are modeled using water-liquid, with a density of 1000 kg/m^3. This configuration ensures that we can accurately capture the droplets’ behavior in the airflow.’

Please see Page 8, lines 226-230 in the revised manuscript.

Furthermore, to support the configuration used in our study, we have provided relevant literature references as listed below. These references also set the droplets as inert particles, and the same physical properties as in our study. In addition, due to the low-humidity nature of the cabin environment during flight, our previous research demonstrated that droplets would rapidly evaporate into a residual [2-4], which is expected to minimize their effect on droplet transport characteristics. However, since this study modeled a large-scale, densely populated cabin environment and considering the high computational resources required, evaporation was temporarily not considered.

[2] Li P, Liu W, Zhang TT. CFD modeling of dynamic airflow and particle transmission in an aircraft lavatory. Building Simulation. 2023;16(8):1375-90.

[3] Ma B, Ruwet V, Corieri P, Theunissen R, Riethmuller M, Darquenne C. CFD simulation and experimental validation of fluid flow and particle transport in a model of alveolated airways. Journal of Aerosol Science. 2009;40(5):403-14.

[4] Park CIP, editor Simulating aerosol movement in experimental chambers using computational fluid dynamics. CSBE/SCGAB 2017 Annual Conference; 2017.

4. Is humidity being considered in this study? If no, please justify thoroughly. As far as reviewer concern, humidity could significantly affect the droplets transportation characteristics.

Response: Thank you for your insightful question regarding the consideration of humidity in our study. We appreciate your concern about the potential impact of humidity on droplet transport characteristics. In this study, we did not explicitly consider humidity as an important factor. Our decision was based on several considerations. Firstly, our previous research has shown that in the low humidity environment of an aircraft cabin, droplets tend to evaporate rapidly after emission. For 10 μm droplets, the evaporation process terminates within 1s. Notably, the smaller the droplet, the faster the evaporation process. This quick evaporation process minimizes the influence of environmental humidity on droplet transport [5-7].

Moreover, the primary focus of our research was to introduce the PCR (Personal Contamination Ratio) method. Given this emphasis, we chose to simplify certain environmental parameters in our simulation phase to highlight the efficacy of the PCR approach.

However, we acknowledge the importance of humidity in this filed study. Additionally, we included the following limitation section in the manuscript. Furthermore, considering that accounting for evaporation effects in the cabin environment would significantly increase computational cost, we have decided, based on our previous studies, to disregard the evaporation of droplets. Rather, we shall concentrate our research on developing the PCR method.

‘In this study, we did not explicitly account for the effects of humidity on droplet transport within the aircraft cabin environment. Our decision was informed by previous research indicating that droplets tend to evaporate rapidly in the extremely dry conditions of an aircraft cabin, thereby minimizing the impact of ambient humidity on droplet dynamics. This simplification was necessary to focus on the primary objective of our research: the development and validation of the Personal Contamination Ratio (PCR) method. While this approach allowed us to highlight the efficacy of the PCR method, we recognize that excluding humidity and evaporation effects may limit the generalizability of our findings to other environments. Future studies should consider more factors to provide a more comprehensive understanding of droplet behavior under varying humidity conditions.’

Please see Page 18, lines 520-529 in the revised manuscript.

[5] Shang Y, Dong J, Tian L, He F, Tu J. An improved numerical model for epidemic transmission and infection risks assessment in indoor environment. Journal of Aerosol Science. 2022 May 1;162:105943.

[6] Li X, Shang Y, Yan Y, Yang L, Tu J. Modelling of evaporation of cough droplets in inhomogeneous humidity fields using the multi-component Eulerian-Lagrangian approach. Building and Environment. 2018 Jan 15;128:68-76.

[7] Li X, Yan Y, Fang X, Tu J. Numerical studies of indoor particulate and gaseous micropollutant transport and its impact on human health in densely-occupied spaces. Environmental Pollution. 2024 Feb 1;342:123031.

5. Authors shall justify why SIMPLE scheme for pressure velocity coupling and second order upwind scheme are chosen. Else, authors might need to find reference (similar study that investigate the effect of droplets dispersion in indoor) to support. Example: https://doi.org/10.1016/j.enbuild.2023.113439

Response: We greatly appreciate your valuable question regarding our choice of the SIMPLE scheme for pressure-velocity coupling and the second-order upwind scheme. Furthermore, it has been demonstrated in earlier research that the second-order upwind scheme and the SIMPLE method for pressure-velocity coupling are especially well-suited for simulations of interior environments.

Following your suggestion, we have revised the description of the SIMPLE scheme as follows:

‘This study employs the SIMPLE scheme for pressure-velocity coupling and the second-order upwind scheme for momentum space discretization. This combination has been proven particularly suitable for indoor environment simulations, aligning well with the scope of our study [8,9].’

Please see Page 7, line 209 and Page 8, lines 210-211 in the revised manuscript.

The authors have carefully read the literature provided by the reviewers and it was found the literature provide valuable insight to our manuscript, and we have added the following references in the manuscript:

[8] Tan H, Othman MHD, Kek HY, Chong WT, Wong SL, Ern GKP, et al. Would sneezing increase the risk of passengers contracting airborne infection? A validated numerical assessment in a public elevator. Energy and Buildings. 2023;297:113439.

[9] Ho X, Ho

---

## [Decision Letter · Decision Letter 2]

19 Aug 2024

Optimizing cabin air inlet velocities and personal risk assessment: introducing the Personal Contamination Ratio (PCR) method for enhanced aircraft cabin infection risk evaluation

PONE-D-24-00989R2

Dear Dr. Shang,

We’re pleased to inform you that your manuscript has been judged scientifically suitable for publication and will be formally accepted for publication once it meets all outstanding technical requirements.

Kind regards,

Krit Pongpirul, MD, MPH, PhD.

Academic Editor

PLOS ONE

Additional Editor Comments (optional):

Reviewers' comments:

Reviewer's Responses to Questions

**Comments to the Author**

1. If the authors have adequately addressed your comments raised in a previous round of review and you feel that this manuscript is now acceptable for publication, you may indicate that here to bypass the “Comments to the Author” section, enter your conflict of interest statement in the “Confidential to Editor” section, and submit your "Accept" recommendation.

Reviewer #4: All comments have been addressed

2. Is the manuscript technically sound, and do the data support the conclusions?

Reviewer #4: Yes

3. Has the statistical analysis been performed appropriately and rigorously? 

Reviewer #4: Yes

4. Have the authors made all data underlying the findings in their manuscript fully available?

Reviewer #4: Yes

5. Is the manuscript presented in an intelligible fashion and written in standard English?

Reviewer #4: No

6. Review Comments to the Author

Reviewer #4: All comments have been addressed well. Please send the manuscript for proofread before publication.

7. PLOS authors have the option to publish the peer review history of their article (what does this mean?). If published, this will include your full peer review and any attached files.

Reviewer #4: No

---

## [Editor Report · Acceptance letter]

27 Aug 2024

PONE-D-24-00989R2 

PLOS ONE

Dear Dr. Shang, 

I'm pleased to inform you that your manuscript has been deemed suitable for publication in PLOS ONE. Congratulations! Your manuscript is now being handed over to our production team.

Kind regards, 

on behalf of

Assoc. Prof. Dr. Krit Pongpirul 

Academic Editor

PLOS ONE